



# Grey water footprint reduction in irrigated crop production: effect of nitrogen application rate, nitrogen form, tillage practice and irrigation strategy

Abebe D. Chukalla[1], Maarten S. Krol[1] and Arjen Y. Hoekstra[1,2]

[1] Twente Water Centre, University of Twente, Enschede, The Netherlands

[2] Institute of Water Policy, Lee Kuan Yew School of Public Policy, National University of Singapore, Singapore

Correspondence to: Abebe D. Chukalla (a.d.chukalla@utwente.nl)

## Abstract

Grey water footprint (WF) reduction is essential given the increasing water pollution associated with food production and the limited assimilation capacity of fresh water. Fertilizer application can contribute significantly to the grey WF as a result of nutrient leaching to groundwater and runoff to streams. The objective of this study is to explore the effect of the nitrogen application rate (from 25 to 300 kg N ha$^{-1}$), nitrogen form (inorganic-N or manure-N), tillage practice (conventional or no-tillage) and irrigation strategy (full or deficit irrigation) on the nitrogen load to groundwater and surface water, crop yield and the grey water footprint of crop production by a systematic model-based assessment. As a case study, we consider irrigated maize grown in Spain on loam soil in a semi-arid environment, whereby we simulate the twenty-years period 1993-2012. The water and nitrogen balances of the soil and plant growth at field scale were simulated with the APEX model. As a reference management package, we assume the use of inorganic-N (nitrate), conventional tillage and full irrigation. For this reference, the grey WF at a usual N application rate of 300 kg N ha$^{-1}$ (with crop yield of 11.1 t ha$^{-1}$) is 1100 m$^3$ t$^{-1}$, which can be reduced by 91% towards 95 m$^3$ t$^{-1}$ when the N application rate is reduced to 50 kg N ha$^{-1}$ (with a yield of 3.7 t ha$^{-1}$). The grey WF can be further reduced to 75 m$^3$ t$^{-1}$ by shifting the management package to manure-N and deficit irrigation (with crop yield of 3.5 t ha$^{-1}$). Although water pollution can thus be reduced dramatically, this comes together with a great yield reduction, and a much lower water productivity (larger green plus blue WF) as well. The overall (green, blue plus grey) WF per tonne is found to be minimal at an N application rate of 150 kg N ha$^{-1}$, with manure, no-tillage and deficit irrigation (with crop yield of 9.3 t ha$^{-1}$). The paper shows that there is a trade-off between grey WF and crop yield, as well as a trade-off between reducing water pollution (grey WF) and water consumption (green and blue WF). Applying manure instead of inorganic-N and deficit instead of full irrigation are measures that reduce both water pollution and water consumption with a 16% loss in yield.

**Key words**: grey water footprint, nitrogen balance, water balance, deficit irrigation, tillage, crop growth, APEX

## 1.    Introduction

Crop yields depend on anthropogenic addition of nitrogen (N), but N fertilizer inevitably result in some N leaching and runoff as well, resulting in the pollution of groundwater and surface water. Freshwater dilutes pollutant loads



entering a water body, which can be interpreted as an appropriation of fresh water (Postel et al., 1996;Falkenmark
and Lindh, 1974;Chapagain et al., 2006;Hoekstra, 2008). The amount of freshwater appropriated to assimilate the
load of pollutants in order to meet ambient water quality standards is called the grey water footprint (WF)
(Hoekstra et al., 2011). For crop production, the grey WF can be expressed as the volume of water per hectare or
per tonne [$m^3$ ha$^{-1}$ or $m^3$ ton$^{-1}$]. Global crop production contributes three quarters to the N-related grey WF in
the world (Mekonnen and Hoekstra, 2015). Anthropogenic N application in agriculture and the resulting
freshwater pollution is expected to increase with the growing production of food, feed, fibre, and biofuel in the
world, driven by population growth and improving living standards. The assimilation capacity of freshwater,
however, is limited, which calls for appropriate management practices that limit the grey WF per tonne of crop
production.
Agricultural management practices that influence the grey WF include the N application rate, the form of N
applied (particularly inorganic-N versus manure or organic-N), and the tillage and irrigation practice. A low N
application rate will hamper plant growth and thus result in a low crop yield (Raun et al., 2002); water pollution
per hectare will be small, but large relative to the volume of crops produced. A high N application rate will result
in a high crop yield, but with high water pollution per hectare and per tonne of crop as well. The reason for the
high water pollution per tonne of crop is that there is a threshold for the N application rate beyond which yield
does not respond (Zhou et al., 2011), while the surplus N contributes to pollution (Carpenter et al., 1998;Vitousek
et al., 2009). The form of N applied is another important factor affecting N losses. Inorganic N is readily available
for uptake by crops (Haynes, 2012), whereas the organic-N contained in manure becomes available only gradually,
as it should first be converted (mineralized) to inorganic form (Ketterings et al., 2005). The mobile nature of
nitrate makes it susceptible for higher risk of leaching (Yanan et al., 1997), while the slow disappearance of
manure makes it susceptible to N losses through runoff before being taken up by the crop (Withers and Lord,
2002). Field operation practices such as tillage affect the water holding capacity of the soil, the movement of
moisture and nutrients in the soil, surface runoff, and eventually crop yield and nutrient load to freshwater. There
are various good reasons why conventional tillage is being practiced: it mixes fertilizer, organic matter and oxygen
in the soil, breaks up surface soil crusts and reduces weeds (Horowitz, 2011). However, conventional tillage
disrupts aggregates within the soil and life cycles of beneficial organisms, increases soil erodability, and results in
soil compaction and tillage pan formation (Triplett and Dick, 2008). Alternatively, no-tillage develops mulch cover,
improves the soil water holding capacity (Dangolani and Narob, 2013) and increases hydraulic conductivity (Azooz
and Arshad, 1996;Triplett and Dick, 2008). The irrigation practice primarily influences the water balance of the
soil, but as a side effect it influences nutrient movement in the soil. The advantage of deficit irrigation compared
to full irrigation is that there may be less leaching and runoff of nutrients (Withers and Lord, 2002), but the
disadvantage is that it may result in reduced N demand as crop growth diminished and reduced N supply as N
transporting agent is reduced and thus reduction in water pollution per unit of crop produced (Gonzalez-Dugo et
al., 2010).



Various studies show how increasing N application rates result in both increased crop yield and N leaching
(Berenguer et al., 2009;Rong and Xuefeng, 2011;Valero et al., 2005;Zhou et al., 2011;Cooper et al., 2012;Good
and Beatty, 2011). Other studies analyse the effect of tillage practices on crop yield (Pittelkow et al., 2015) or the
effect of tillage practices and N fertilizer forms on crop yield (Yu et al., 2016) or the effect of manure versus
inorganic N fertilizer application on nitrate leaching (Huang et al., 2017;Yanan et al., 1997) or the effect of
different tillage practices and N application rates on yield and N leaching (Huang et al., 2015). There are quite
some studies also on the relation between rates of irrigation and N application and crop yield (Yin et al., 2014;Al-
Kaisi and Yin, 2003;Rimski-Korsakov et al., 2009). These earlier studies provide insight in the effects of individual
management practices on yield, water productivity, or leaching, however most of the studies vary only one or
two management practices, not considering the combined effect of N application rate, N form, tillage practice
and irrigation strategy. Besides, none of these studies consider the effect on the pollutant load per unit of crop
obtained or the effect on the grey WF per tonne.
It is challenging to conduct field experimental studies and even more laborious and expensive to study the effects
of a comprehensive list of different combinations of management practices. Besides, leaching and runoff of N
from fields is difficult to determine through field experiments; N that can be measured in groundwater and
streams originates from different sources and cannot easily be attributed to an experimental field. An alternative
approach avoiding these downsides is to use modelling (Chukalla et al., 2015;Ragab, 2015).
The objective of this study is to explore the effect of nitrogen application rate, nitrogen form, tillage practice and
irrigation strategy on the nitrogen load to groundwater and surface water, crop yield and the grey water footprint
of crop production by a systematic model-based assessment. We apply the Agricultural Policy Environmental
eXtender (APEX) model, which simulates nutrient and water balances of the soil and plant growth, is able to
simulate the effect of a wide variety of agricultural management practices, and has been applied for a wide variety
of cases (Wang et al., 2012;Gassman et al., 2010;Liu et al., 2016;Clarke et al., 2017;Chen et al., 2017). As a case
study, we consider irrigated maize grown in Badajoz in Spain on loam soil in a semi-arid environment, whereby
we simulate the twenty-years period 1993-2012.
The method to estimate grey WFs in the current study is more advanced than in previous studies. (Franke et al.,
2013) distinguish three tiers to estimate grey WFs from diffuse pollution. The tier-1 approach is based on expert-
based assumptions on which fractions of applied or surplus N in the soil will leach or run off given contextual
factors. It provides a first rough estimate of the N load without describing the interaction and transformation of
different chemical substances in the soil or along its flow pathways (see for instance Mekonnen and Hoekstra
(2011), and Brueck and Lammel (2016)). The more advanced tier-2 approach for estimating grey WFs from diffuse
pollution is based on an N balance approach, applying a simplified model approach (see for example Mekonnen
and Hoekstra (2015), and Liu et al. (2012)). The current study is the first one to apply the tier-3 approach, which
explicitly considers physical and biochemical processes using an advanced water and nutrient balance model (the
APEX model). As an additional component of the current study, we will compare the N leaching-runoff fractions



that result from the APEX simulations with the leaching-runoff fractions estimated with the simpler tier-1
approach, in order to find out the added value of employing the advanced model approach.

## 2.   Method and data
### 2.1.   Modelling the soil water & nitrogen balances and crop growth

The effect of various combinations of management practices on water flows (like soil evaporation, crop
transpiration, percolation and runoff), N flows (like N uptake by plants, leaching and runoff) and crop growth are
simulated using the APEX model, a dynamic, deterministic and process-based model with a daily time step
(Williams and Izaurralde, 2006). Below we briefly summarise the processes simulated in the model. More detailed
descriptions of the processes and the equations to simulate these processes can be found in the documentation
of APEX (Williams et al., 2008).

The water balance component of APEX encompasses key processes that impact the soil water compartment in
the hydrologic cycle. Initially, incoming inputs such as precipitation, snowmelt, or irrigation is partitioned between
surface runoff and infiltration. Surface runoff volume is simulated using a modified Soil Conservation Service curve
number technique described by Williams (1995). Infiltrated water can be stored in the soil profile, be lost via
evapotranspiration (ET), percolate vertically to groundwater, or flow laterally as subsurface flow, with a quick and
slow component. Reference ET is calculated using the Penman-Monteith method. The actual ET, an important
variable in estimating green and blue WF of crop production, is computed by simulating evaporation from the soil
and transpiration from plants separately, considering the soil moisture status and how agricultural management
practices affect the root zone. Percolation and lateral flow are computed using storage routing and pipe flow
equations described by Gassman et al. (2010). A deep groundwater table is assumed and thus capillary rise, which
APEX would simulate using storage routing (Gassman et al., 2010), is not considered in the water balance.

The N balance of the soil in APEX is computed based on inputs and outputs and conversion processes (Figure 1).
N is added to the soil-plant system through natural and anthropogenic pathways. Natural N inputs include wet
and dry deposition (Anderson and Downing, 2006) and N fixation, through lightning and through biological
fixation by legume plants (Carpenter et al., 1998). Anthropogenic input occurs when inorganic or organic N
fertilizers are applied (Vitousek et al., 2009). N outputs include N uptake by crops (partly harvested and removed
later on), denitrification, volatilization, nitrate-N losses through leaching, horizontal losses of organic N with
eroded sediments, and horizontal losses of inorganic N through surface runoff, or lateral subsurface flow. N
transformation includes mineralization, immobilization and nitrification.





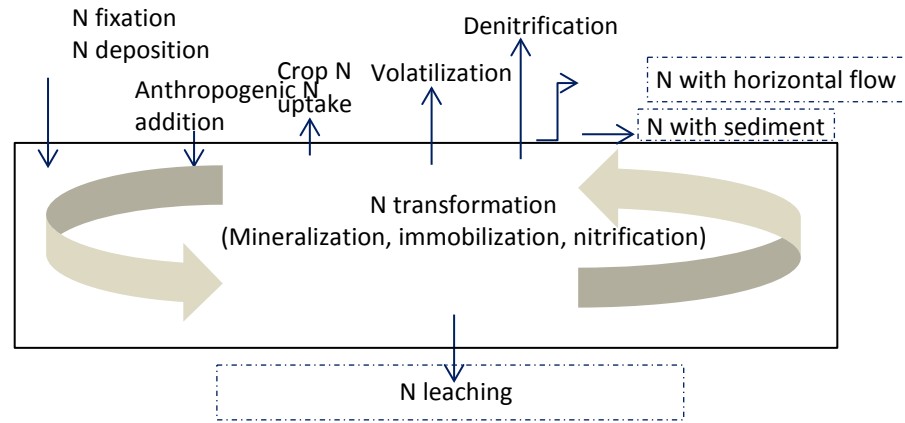

**Figure 1**. Nitrogen fluxes into and from the root zone, and N transformation.

APEX simulates the growth of annual and perennial crops based on the EPIC model (Williams et al., 1989), an
energy-driven crop growth model using a radiation-efficiency approach to simulate the generation of biomass.
Potential biomass production is derived as function of leaf area index and climatic variables (solar radiation, $CO_2$,
air humidity and temperature). Phonological development of the crop is based on heat unit accumulation. Annual
crops grow from planting date to harvest date or until the accumulated heat units equal the potential heat units
for the crop (Steduto, 1997). Daily potential growth is lowered to actual growth using the most limiting stress
factor, considering stresses caused by water, nutrients (N and P), temperature and aeration, which are evaluated
by assigning stress factors (from 0, high stress, to 1, no stress). Root growth is constrained based on the most
limiting stress caused by soil strength and temperature. Total biomass is partitioned to root and above ground
biomass, and from the above-ground biomass is the economic yield is partitioned using harvest index.

## 2.2.  The grey water footprint of growing crops

The grey water footprint (WF), an indicator of appropriated pollution assimilation capacity, is calculated following
the Global Water Footprint Standard (Hoekstra et al., 2011), which means that the total pollutant load entering
freshwater (groundwater or surface water) is divided by the difference between the maximum acceptable
concentration for that pollutant and the natural background concentration for that pollutant. The grey WF can
be expressed in two different ways, either as a water volume per ha, or as a water volume per tonne of crop:

$$Grey\ WF\ per\ hectare\ = \frac{L}{C_{max} - C_{nat}}\ \left[m^3\ ha^{-1}\ y^{-1}\right] \qquad (1a)$$

$$Grey\ WF\ per\ tonne = \frac{Grey\ WF\ per\ hectare}{Y}\ \left[m^3\ t^{-1}\right] \qquad (1b)$$





where L (kg ha$^{-1}$ y$^{-1}$) is the pollutant load to surface water and groundwater, $C_{max}$ and $C_{nat}$ are the maximum
acceptable and natural concentrations (kg m$^{-3}$), and Y the crop yield (t ha$^{-1}$ y$^{-1}$).
The total N load to freshwater (L, in kg N ha$^{-1}$ y$^{-1}$) is calculated as the sum of the N load in surface runoff, the N in
quick subsurface flow, the N in slow subsurface flow, the N adsorbed to eroded sediments and the N in
percolation. Each of these N loads are simulated separately in APEX.
A maximum acceptable N concentration of 50 mg nitrate-N L$^{-1}$ (or 11.3 mg N L$^{-1}$) is adopted, based on the EU
Nitrates Directive (Monteny, 2001). The natural concentration was considered to be 0.5 mg N L$^{-1}$, following for
example (de Miguel et al., 2015).
Next to the grey WF, the green and blue WF of crop production are calculated as well, again using the Global WF
standard (Hoekstra et al., 2011). The green WF refers to the rainwater consumed (water evaporated or
incorporated into the crop), while the blue WF refers to the irrigation water consumed (which comes from surface
water or groundwater). Together, the green and blue WF are called the consumptive WF. The consumptive WF
per tonne of crop is calculated by dividing the ET over the growing period by the crop yield.

## 2.3.  Leaching-runoff fraction

As an additional component of the current study, we will compare the N leaching-runoff fraction simulated
through APEX (tier-3 level estimation) with the leaching-runoff fraction estimated with the simpler estimation
approach (tier-1) as applied in previous studies, in order to find out when the simple tier-1 approach suffices and
when it doesn't.
The leaching-runoff fraction can be defined in two ways (Franke et al., 2013). In the first definition, the leaching-
runoff fraction, called α, is defined as the percentage of the amount of chemical applied to the field as fertilizer
that is lost to groundwater through leaching or to surface water through runoff. In the second definition, the
leaching-runoff fraction, now called β, is defined as the percentage of the amount of 'surplus chemical' in the soil
that is lost to groundwater through leaching or to surface water through runoff. The 'surplus chemical' in the soil
is defined as the amount of chemical applied minus the uptake of the chemical by the crop.
$$\alpha = \frac{L}{Appl} \qquad\qquad (2)$$
$$\beta = \frac{L}{Surplus} \qquad\qquad (3)$$
where α and β are the leaching-runoff fractions, and where L (kg N ha$^{-1}$ y$^{-1}$) is the N load to freshwater bodies,
Appl (kg N ha$^{-1}$ y$^{-1}$) the N fertilizer applied, and Surplus (kg N ha$^{-1}$ y$^{-1}$) the N applied but not taken up by the plant.




At the tier-3 level, the fractions α and β are not used in the calculations, but they can easily be calculated
afterwards, based on the outputs of the model. At the tier-1 level, α and β can be estimated following the
guidelines of Franke et al. (2013). According to these guidelines, the leaching-runoff fractions lie between a
minimum and a maximum value (0.01 to 0.25 for α and 0.08 to 0.8 for β). The precise value is estimated based
context-specific environmental and management factors, using the following equations:

$\alpha = \alpha_{min} + \left[\frac{\sum_i s_i * w_i}{\sum_i w_i}\right] * (\alpha_{max} - \alpha_{min})$ (4)
$\beta = \beta_{min} + \left[\frac{\sum_i s_i * w_i}{\sum_i w_i}\right] * (\beta_{max} - \beta_{min})$ (5)

where $s_i$ is score for the leaching runoff potential for environmental or management factor i and $w_i$ is the weight
of that factor.

## 2.4. Simulation set-up

We carry out model simulations with APEX for 56 management packages, whereby each management package
consists of a certain combination of management practices. We consider all possible combinations of seven N
application rates, two N forms, two tillage practices, and two irrigation strategies (Table 1). As a reference
management package, we assume the use of inorganic N fertilizer (nitrate) in combination with conventional
tillage and full irrigation. Conventional tillage is the most wide-spread tillage practice in the EU (EUROSTAT, 2013)
and full irrigation is the most common irrigation practice, aimed at achieving maximum yield.

**Table 1.** Research set-up: the APEX model is used to simulate the effect of 56 management packages
(combinations of different management practices) on ET, crop yield, nitrogen load to freshwater, and green, blue
and grey WF.

| Management practices | Modelling | Effects |
|---|---|---|
| • Nitrogen application rates: 25, 50, 100, 150, 200, 250 or 300 kg N ha$^{-1}$ <br> • Nitrogen forms: inorganic-N (nitrate) or organic-N (manure) <br> • Tillage practices: no-tillage or conventional tillage <br> • Irrigation strategies: full or deficit irrigation | Soil water & nutrient balances and crop growth model (APEX) | - ET <br> - Yield <br> - N load <br> - Green, blue, grey WF |


The EU Nitrate Directive legally restricts annual farm application of manure in EU member states to 170 kg N ha$^{-1}$
$^{-1}$ y$^{-1}$, or in case of derogation up to 250 kg N ha$^{-1}$ (Amery and Schoumans, 2014;Van Grinsven et al., 2012). Surveys
in Spain, however, show that application rates of 300-350 kg N ha$^{-1}$ y$^{-1}$ are common to cultivate maize in the Ebro





Valley (Berenguer et al., 2009) and up to 300 kg N ha⁻¹ in La Mancha (Valero et al., 2005). As the upper value for
the N application rate in our simulations we apply 300 kg N ha⁻¹.
The fertilization is assumed to be performed in two splits (30% in a first round, at planting for mineral fertilizer
and 15 days before planting for manure; 70% in a second round, one month after planting). In the first round of
application, inorganic fertilizer is assumed to be nitrate-N and applied through broadcasting while manure is
assumed to be injected. Manure injection is getting recognition in the EU and in the world due to its many
advantages, including reduction of N losses to freshwater and to the atmosphere and bad odour (Van Dijk et al.,
2015;van den Pol-van Dasselaar et al., 2015). In the second round, both the manure and nitrate-N fertilizers are
added as side-dressing.
As for the inorganic N applied, we assume that the N is 100% in the form of nitrate. Manure is generally contained
of mostly organic N, and a smaller amount of inorganic N (Ketterings et al., 2005;Pratt and Castellanos, 1981). In
this study, we assume the manure composition as in the APEX database: 91.67% organic N, 8.33% inorganic N
(0.23% nitrate and 8.10% ammonium N). In addition, the current study assumes that other nutrients (P, K and
micro nutrients) do not to constrain crop production.
We simulate conventional tillage in APEX as two times ploughing to a depth of 20 cm at thirty and fifteen days
before sowing date and one time harrowing following the emergence of the seed. The two times ploughing is the
average of what is most common, namely one to three times tilling (Nagy and Rátonyi, 2013;FAO, 2016). With
the tillage depth of 20 cm we follow the average estimate reported by Townsend et al. (2015) and FAO (2016).
No-tillage, a form of conservation tillage that is strongly encouraged by the EU agricultural policy (De Vita et al.,
2007), is simulated as no soil disturbance; the stubble of the previous crop is kept on the field.
We simulate full irrigation in APEX by irrigating up to field capacity as soon as the soil water content would
otherwise drop below a level at which water stress occurs. Deficit irrigation is simulated to allow for 20% plant
water stress, a deficit level that can achieve 61-100% of full ET (Fereres and Soriano, 2007). With this irrigation
strategy, average water productivity is higher than in case of full irrigation (Chukalla et al., 2015). We assume the
use of furrow irrigation, the irrigation technique that covered the largest irrigated area in the EU in 2010,
particularly in the Eastern and Mediterranean regions of Europe (EUROSTAT, 2016).
## 2.5. Data
The following climatic and soil data have been collected for Badajoz in Spain (38.88⁰ N, -6.83⁰ E; 185 m above
mean sea level). Daily observed rainfall and temperature data (for the period 1993-2012) are extracted from the
European Climate Assessment and Dataset (Klein Tank et al., 2002). These data have been subject to homogeneity
testing and missing data have been filled with observations from nearby stations (Klein Tank, 2007). Mean





monthly wind speed data are taken from the FAO CLIMAWAT database (Smith, 1993). Daily reference
evapotranspiration is calculated using the Penman-Montheith equation, as implemented in APEX (Williams et al.,
2008). The physical and chemical characteristics of the soil, and nutrient content in the soil (nitrogen, phosphorus,
carbon) that are used in APEX are extracted from the 1×1 km² resolution European Soil Database (Hannam et al.,
2009). Using the Soil Texture Triangle Hydraulic Properties Calculator from (Saxton et al., 1986), we identified the
soil at our location as loam soil. We use a soil albedo of 0.13 for a loam soil at its field capacity (Sumner, 1999).
Regarding crop parameters, we use the default values from the APEX model. We simulate zero pest stresses (from
insects and diseases) to crop growth.
Soil moisture content is initialised using the standard procedure in APEX, which is based on average annual rainfall
within the period considered (1993-2012). We adjust initial organic-N content for each simulation so that the N
build-up in the soil over the 20-year period is zero. We apply the graphical time-series inspection method
(Robinson, 2002) to determine the warm-up period, i.e. the period in which simulation results are still affected
by the model initialization. We find that we best exclude the first five years of the simulation, thus we show results
for the period 1998-2012.

## 3.  Results

### 3.1.  Pollutant loads and grey WF for the reference management package

N out-fluxes from the soil for maize production under the reference management package (inorganic-N,
conventional tillage, full irrigation) for different N application rates are shown in Figure 2. The N out-fluxes are
denitrification and volatilization to the atmosphere, N harvested with the crop, and N loads to freshwater adhered
to sediment and dissolved in percolation and runoff. All of these N out-fluxes increase with the N application rate
and with the N surplus in the root zone (N application minus crop uptake). For all N application rates the N
harvested with the crop is the main share of the N out-flux. For larger N application rates, the share of N leaching
increases substantially. For all application rates, N leaching to groundwater constitutes at least 95% of the total
N load to freshwater, and the N flux to surface water (N dissolved in runoff plus N in eroded sediments) 5% at
most.
Crop yields increase with the N application rate as a result of reduced N stress. Yields stabilize at larger N
application rates. The yield increase, however, comes at a price: the N load to freshwater, through leaching, runoff
and eroded sediment, increases exponentially. As a result, large N-application rates result in a large grey WF
(Figure 3). At lower N-application rates, crop yields decline as a consequence of N stress. While the grey WF in $m^3$
$ha^{-1}$ keeps on declining with lower N-application rates, the grey WF in $m^3$ $t^{-1}$ starts increasing again at very low N-
application rate (in our case when the N-application rate drops below 50 kg N $ha^{-1}$. The smallest grey WF per
tonne can be found at an N-application rate of 50 kg N $ha^{-1}$, where yield is substantially lower than the maximum,




but where additional N application goes along with increasing N load per unit of crop yield gain, thus with
increasing grey WF per tonne.

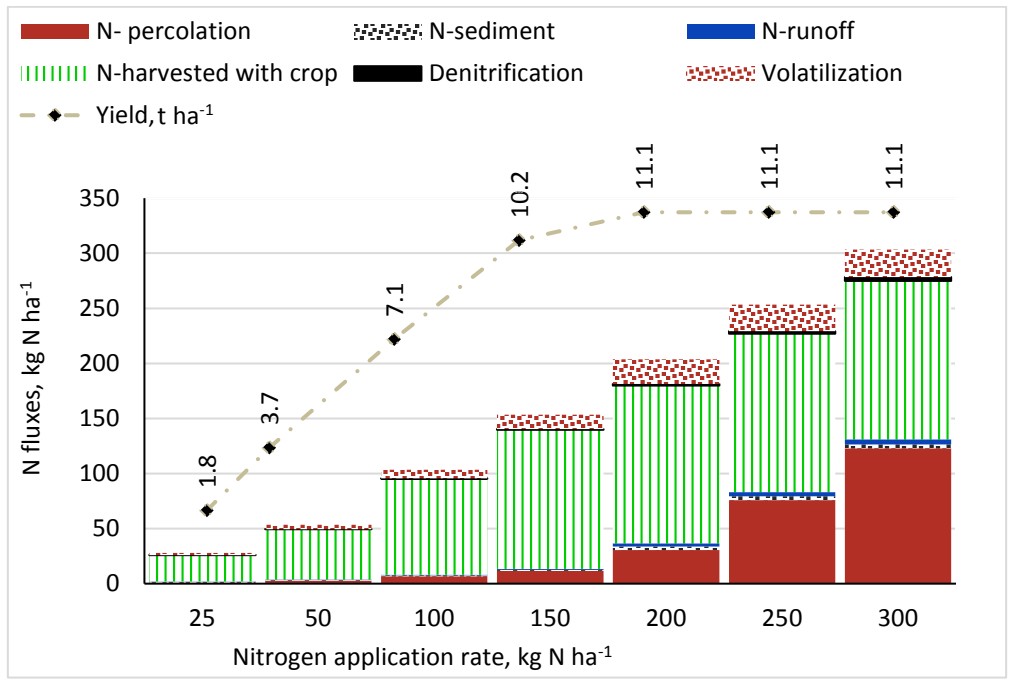

**Figure 2**. Nitrogen out-fluxes and yield for an irrigated maize field for a range of N-application rates under the
reference management package (inorganic-N, conventional tillage, full irrigation).

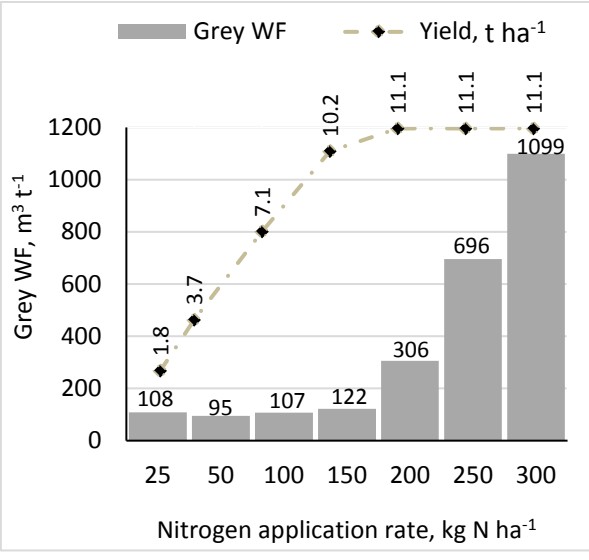
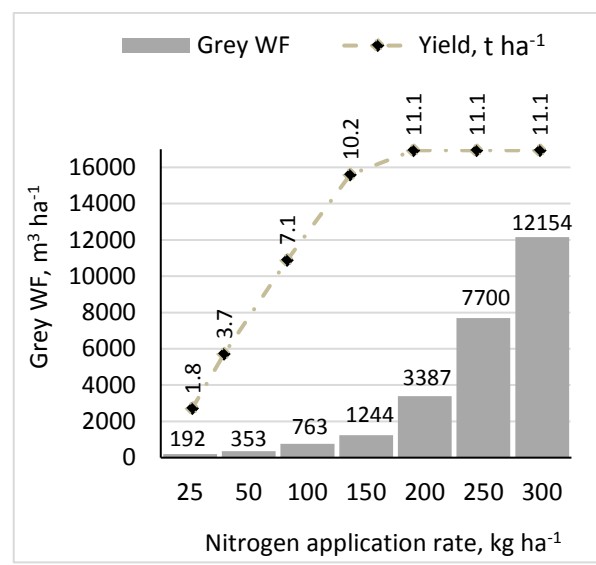




**Figure 3**. Grey WF of maize production in $m^3$ $t^{-1}$ (left) and $m^3$ $ha^{-1}$ (right) for a range of N-application rates under the reference management package.

### 3.2. Effect of fertilizer form, tillage practice and irrigation strategy on grey WF

Figure 4 shows that, at a given N-application rate, the grey WF in $m^3$ $t^{-1}$ can be higher or lower than for reference management package, by changing to manure, no-tillage or deficit irrigation, or a combination of those. Across the whole range of N application rates, the use of manure results in a smaller grey WF per tonne than the use of nitrate fertilizer. The effect of the tillage practice and irrigation strategy on the grey WF depends on the N-application rate. We can identify three ranges for the application rate, each with a different management package resulting in the smallest grey WF per tonne:

I. Application rates up to 125 kg N $ha^{-1}$: the grey WF is smallest for manure with conventional tillage and deficit irrigation;

II. Application rates between 125 and 225 kg N $ha^{-1}$: the grey WF is smallest for manure with conventional tillage and full irrigation;

III. Application rates above 225 kg N $ha^{-1}$: the grey WF is smallest for manure with no-tillage and full irrigation.

At low and intermediate N-application rates (ranges I-II), the advantage of conventional tillage over no-tillage is that it decreases the hydraulic conductivity of the soil (because of the removal of fine cracks in the soil), which reduces percolation and thus N leaching. At high N-application rates (range III), no-tillage appears to be better. The disadvantage of increased hydraulic conductivity is now compensated by another effect: no-tillage results in improved soil texture: the soil remains intact, which in combination with the build-up of organic content creates favourable conditions for soil organisms that help to glue the soil particles and increase the number of micro-pores and macro-pores in the soil. This increases the soil water holding capacity and thus N holding capacity of the soil, resulting in lower N leaching (by 30%) and higher yield (by 3.6%).

At low application rates (ranges I), deficit irrigation decreases the amount of water available for percolation and thus reduces N leaching as well. At intermediate and higher N-application rates (ranges II-III), full irrigation has a smaller grey WF per tonne as compared to deficit irrigation because of the higher crop yield. With the absence of water stress and the higher yield, the N uptake by the crop is higher, resulting in a lower N surplus in the root zone and decreased N leaching.




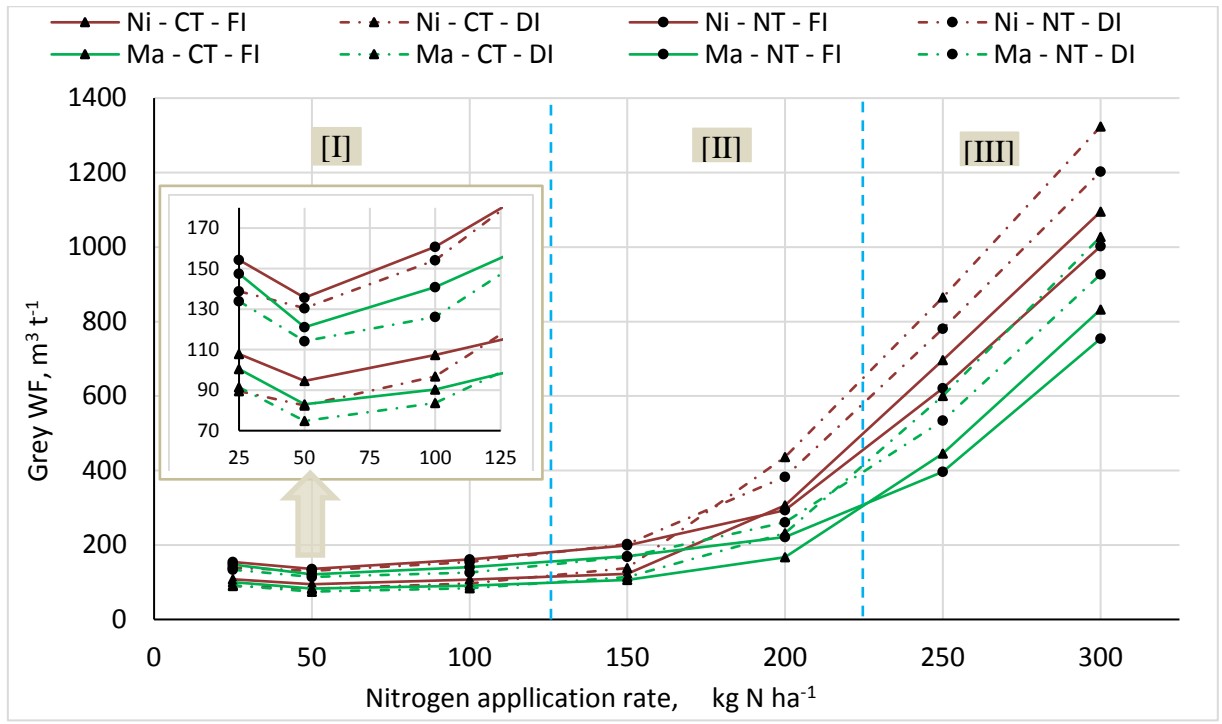

**Figure 4**. The effect of N application rate, N form, tillage practice and irrigation strategy on grey WF per tonne. Considering which management package gives the lowest grey WF, three ranges can be distinguished: [I] N application rates up to 125 kg N ha$^{-1}$, [II] N application rates between 125 and 225 kg N ha$^{-1}$, [III] N application rates above 225 kg N ha$^{-1}$. Red lines refer to nitrate (Ni); green lines refer to manure (Ma). Circular markers refer to no-tillage (NT); triangular markers refer to conventional tillage. Dashed lines refer to deficit irrigation (DI); solid lines refer to full irrigation (FI).

The smallest grey WFs per tonne are found for an N application rate of 50 kg N ha$^{-1}$. Taking the reference management package with an N application rate of 300 kg N ha$^{-1}$ as a starting point, one can reduce the grey WF per tonne of crop production by reducing the N application rate while keeping the management package fixed, by shifting the management package to one with a smaller grey WF, or both (Table A.1 in Appendix). Reducing the N application rate from 300 kg N ha$^{-1}$ to the optimum of 50 kg N ha$^{-1}$ under the reference management package will reduce the grey WF by 91% (from around 1100 to 95 m$^3$ t$^{-1}$), but the crop yield will reduce by two thirds (from 11.1 to 3.7 t ha$^{-1}$). When, at the application rate of 50 kg N ha$^{-1}$, shifting from the reference management package to organic N and deficit irrigation, once can further reduce the grey WF by 21% (from around 95 to 75 m$^3$ t$^{-1}$), with a yield reduction of 5% (from 3.7 to 3.5 t ha$^{-1}$).

### 3.3. Reducing grey WF vs consumptive WF





Both ET and yield increase with increasing N application rate, but level off at large N application rates (Figure 5a).
Adding more N at relatively low application rates has a larger impact on Y increase than on ET increase. As a result,
the consumptive WF per tonne, defined as ET over Y, decreases with increasing N application rate, levelling off at
larger N application rate (Figure 5b). The grey WF per tonne, however, exponentially increases with increasing N
application rate. As a result, the sum of grey and consumptive WF has a minimum somewhere at intermediate N
application rate, at 150 N ha$^{-1}$ in the case of our reference management package. The total WF is dominated by
the consumptive WF for smaller N application rates and by the grey WF for larger N application rates.

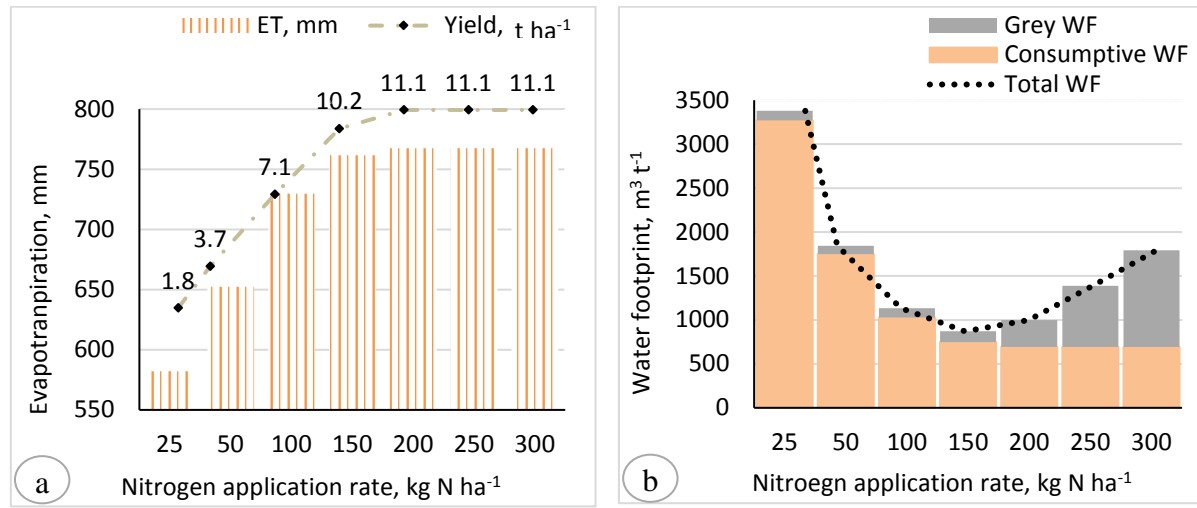

**Figure 5.** Evapotranspiration and yield (Fagard et al.) and consumptive WF and grey WF per tonne (b) for the
reference management package.

Figure 6 shows the total (grey+consumptive) WF per tonne for the reference management package for different
N application rates (the solid red line). For each given N application rate, shifting to another management package
can reduce the total WF. At N application rates of 25, 50 and 100 kg N ha$^{-1}$, the total WF can be reduced by shifting
towards no-tillage and deficit irrigation. At N application rates of 150 kg N ha$^{-1}$, the total WF can be reduced by
shifting towards organic N, no-tillage and deficit irrigation. Finally, at N application rates of 200, 250 and 300 kg
N ha$^{-1}$, the total WF can be reduced by shifting towards organic N and no-tillage. The total WF reductions shown
in the figure are the net effect of changes in the consumptive WF and grey WF; in some cases the total WF
decrease is at the cost of some grey WF increase.




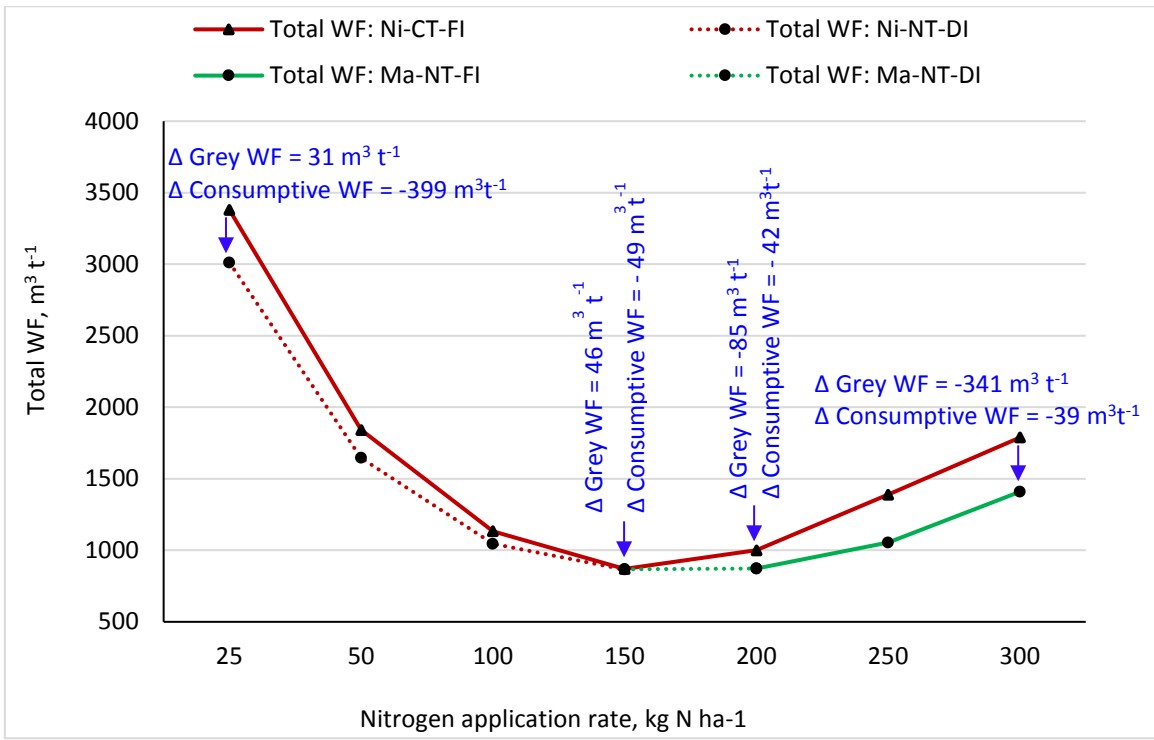

**Figure 6.** The total (green, blue plus grey) WF per tonne for the reference management package and for a management package with the largest total WF reduction potential. Red lines refer to nitrate (Ni); green lines refer to manure (Ma). Circular markers refer to no-tillage (NT); triangular markers refer to conventional tillage. Dashed lines refer to deficit irrigation (DI); solid lines refer to full irrigation (FI).

### 3.4. Resultant leaching-runoff fractions

The N leaching-runoff fractions $\alpha$ and $\beta$ for different N application rates for the reference management package, as calculated here with the tier-3 approach, are shown in Figure 7. The $\alpha$ values, which show the ratio of the N load to fresh water to the N application rate are lower than the $\beta$ values, which show the ratio of the N load to the N surplus in the soil. This can be logically understood, because the N load to freshwater (in the numerator of both ratios) is the same, while the $\alpha$ ratio has the total N application rate in the denominator, while the $\beta$ ratio has the relatively smaller N surplus (which is only a fraction of the N applied) in the denominator.

With increasing N application rate, both N surplus in the soil and the N load to freshwater increase exponentially (Figure 2). The $\alpha$ values grow with increasing N application rate, because the N load to freshwater increases quicker with increasing N application rates than the application rate itself. The $\beta$ values also grow with increasing N application rates, because denitrification and volatilization do not grow proportionally to the growth in N surplus, which leads to greater fractions of the surplus getting lost through leaching and runoff.





413

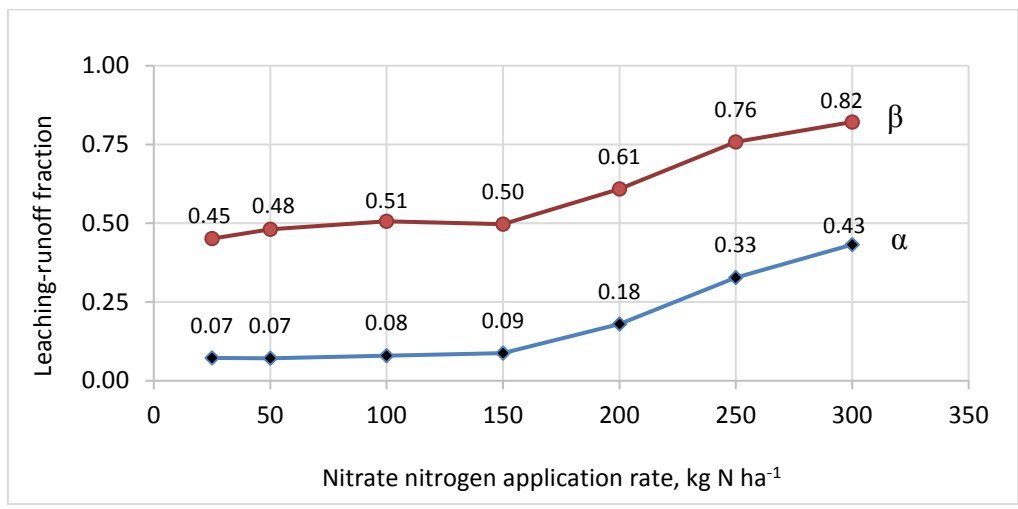

414

**Figure 7.** The N leaching-runoff fractions α and β calculated per N application rate for the reference
management package.

Figures 8 and 9 show α and β values for different management packages and N application rates. For comparison,
the figures also show the α and β values when estimated based on the simpler tier-1 approach (Tables A.2 and
A.3 in Appendix), which estimates α and β within minimum and maximum values based on context-specific
environmental and management factors (see section 2.3). The calculated leaching-runoff fractions based on the
APEX model (tier-3 approach) for all management packages across the range of N application rates fall within the
range set by the minimum and maximum leaching-runoff fractions margins as applied in the tier-1 approach
(Franke et al., 2013), except for α for very high N application rates.

For N applications rates in the range up to 150 kg ha$^{-1}$, the tier-1 approach gives a good proxy for the α value. For
the reference management package, the most common practice, the tier-1 approach even yields nearly the same
α values as the more advanced tier-3 approach. For N applications rates exceeding about 150 kg ha$^{-1}$, the tier-1
approach underestimates the leaching-runoff fraction and thus the grey WF. The β values estimated based on the
tier-1 approach are comparable to the ones calculated at the tier-3 level for the management packages with
manure and conventional tillage. For the other management packages, β is underestimated with the tier-1
approach. Also for N application rates of 250 kg ha$^{-1}$ and beyond, the tier-1 approach underestimates β.

The leaching-runoff fractions from the application of inorganic N (nitrate) calculated at the tier-3 level are larger
than these for organic N (manure), a distinction that is not made in the tier-1 approach.





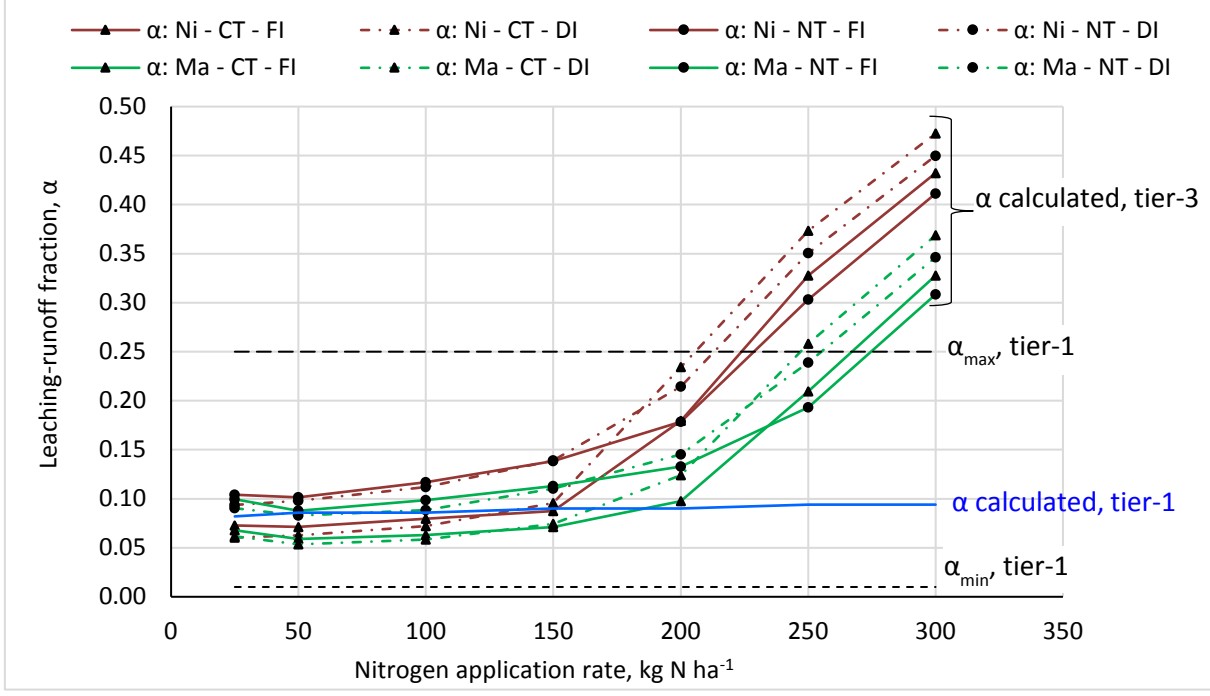

**Figure 8.** N leaching-runoff fractions α for different management packages and N application rates following from the tier-1 or tier-3 approach. Red lines refer to nitrate (Ni); green lines refer to manure (Ma). Circular markers refer to no-tillage (NT); triangular markers refer to conventional tillage. Dashed lines refer to deficit irrigation (DI); solid lines refer to full irrigation (FI).




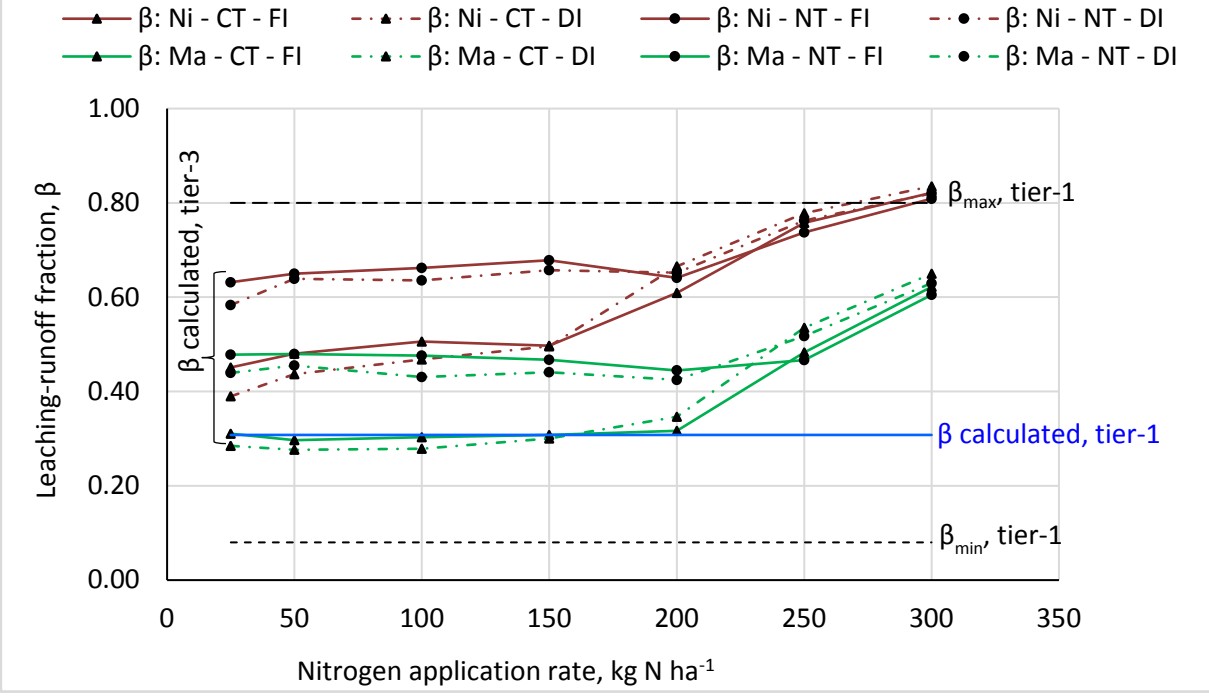

**Figure 9.** N leaching-runoff fractions β for different management packages and N application rates following from the tier-1 or tier-3 approach. Red lines refer to nitrate (Ni); green lines refer to manure (Ma). Circular markers refer to no-tillage (NT); triangular markers refer to conventional tillage. Dashed lines refer to deficit irrigation (DI); solid lines refer to full irrigation (FI).



## 4.   Discussion

The study shows that there is not one combination of management practices that minimises grey WF or overall WF and maximises crop yield at the same time. Table 2 shows that the best combination of practices depends on what variable is optimised. Yield is optimal when there is neither nitrogen stress nor water stress, so at high N application rate and full irrigation. The highest yield (11.5 t/ha) is found for when N is applied in the form of manure and the case of no-tillage. The total WF per tonne (the sum of the green, blue and grey WF) is smallest at 150 kg N ha$^{-1}$, manure application, no-tillage and deficit irrigation. The yield in this case, 9.3 t/ha, is below-optimum. There is both nitrogen and water stress, but the latter is more important. The grey WF per tonne is smallest at 50 kg N ha$^{-1}$, manure application, conventional tillage and deficit irrigation. This, however, reduces the yield to 3.5 t/ha because of nitrogen stress. Deficit irrigation gives some water stress as well, but at such high nitrogen stress, it is the latter that constrains crop yield. Our results confirm the finding by (Mekonnen and Hoekstra, 2014) that there is a trade-off between consumptive WF per tonne and grey WF per tonne, i.e. a trade-off between reducing water consumption  and water pollution.

**Table 2**. The measures that give the optimum grey WF per tonne, total WF per tonne, or yield.

| Indicator / Management practice | Highest yield In t ha$^{-1}$ | Smallest total WF* in m$^3$ t$^{-1}$ | Smallest grey WF in m$^3$ t$^{-1}$ |
|---|---|---|---|
| Nitrogen application rate | 200 kg N ha$^{-1}$ | 150 kg N ha$^{-1}$ | 50 kg N ha$^{-1}$ |
| Nitrogen form | Manure | Manure | Manure |
| Tillage practice | No-tillage | No-tillage | Conventional tillage |
| Irrigation strategy | Full irrigation | Deficit irrigation | Deficit irrigation |

* Total WF refers to the sum of the green, blue and grey WF.

The response of maize yield to nitrogen input as simulated in this study with the APEX model is comparable with the shape of the N-response curves for a few crops, including maize, constructed for the EU based on field measurements from various earlier studies (Godard et al., 2008). Our finding is also consistent with the results presented by Berenguer et al. (2009), who carried out field experiments for maize for similar conditions in Spain (Figure 10). For every given N input, their yields are a bit higher than from our study, which may relate to the fact that  Berenguer et al. (2009) used a high-yield maize variety.



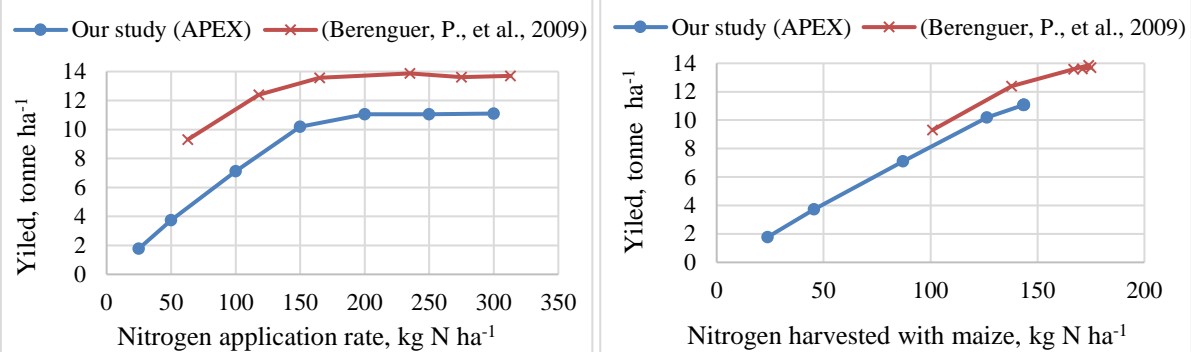

**Figure 10.** The maize yield simulated in our study in relation to N application rate (left) and N harvested with maize (right) in comparison to the maize yields from field experiments by Berenguer et al. (2009) when corrected for zero N build-up in the root zone.

An inter-model comparison for the case of no N stress and no water stress (taking optimal N application rate and full irrigation) for exactly the same growing conditions in Spain shows similar crop yields and net irrigation supply. The current study, using the APEX model, simulates a net irrigation supply of 638 mm and a maize yield of 11.1 t ha$^{-1}$, while in an earlier study, employing the AquaCrop model (Steduto et al., 2011), we simulate an irrigation supply of 630 mm and a maize yield of 11.9 t ha$^{-1}$ (Chukalla et al., 2015).

Simulated yields, N loads to freshwater and grey WFs under different management packages are subject to the local environmental conditions of our case in Spain, which means that they cannot simply be transferred to other conditions. Besides, even for our specific case, the outcomes are subject to uncertainties inherent to any modelling effort (Kersebaum et al. (2016). We have also excluded other factors relevant in crop production, like the effects of weeds, pests and diseases. Therefore, the precise values presented should be taken with caution; the value of our study rather lies in the understanding it provides on how different agricultural management practices can affect yield, N load and resultant grey WF of crop production, and how and why there are inevitable trade-offs between crop yield, water consumption and water pollution.

## 5. Conclusion

This paper provides the first detailed study on potential grey WF reduction of growing a crop by analysing the effect of a large number of combinations of different management practices. The paper shows that, when choosing a certain N application rate and when choosing between inorganic versus organic fertilizer, between conventional versus no tillage, and between full versus deficit irrigation, two inevitable trade-offs are made. The first trade-off is between crop yield and water pollution (grey WF). Whereas maximizing crop yields requires a relatively high N application rate and full irrigation, minimizing water pollution per unit of crop requires deficit irrigation and seeking a balance between N application rate (and associated water pollution) and the resultant





yield. The second trade-off is between reducing water pollution (grey WF) and water consumption (green and blue WF). Minimizing consumptive water use per tonne requires a higher N application rate (150 kg N ha$^{-1}$ in our case) than minimizing water pollution per tonne (50 kg N ha$^{-1}$ in our case). Applying manure instead of inorganic-N and deficit instead of full irrigation are measures that reduce both water pollution and water consumption per tonne. However, for minimizing water pollution per tonne one can better choose for conventional tillage, because that reduces leaching, whereas for minimizing water consumption per tonne the no-tillage practice is to be preferred, because that reduces soil evaporation.

The study gives some support to the simple tier-1 approach of estimating the grey WF of applying N fertilizer as proposed by Franke et al. (2013), but only for N application rates below 150 kg ha$^{-1}$. Below that, the $\alpha$ value is estimated in the proper range (in our specific case), but the $\beta$ value is underestimated. Beyond the N application rate of 150 kg ha$^{-1}$, the tier-1 approach underestimates the leaching-runoff fraction, by not accounting for the fact that N uptake by the crop is stabilizing and that denitrification and volatilization don't increase proportionally with growing N inputs, which results into an increasing fraction of the N surplus in the soil lost through leaching, runoff and erosion.

## Acknowledgments

This research was conducted as part of the EU-FP7-funded project FIGARO. The paper was partially developed within the framework of the Panta Rhei Research Initiative of the International Association of Hydrological Sciences (IAHS).

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

## Appendix

Table A.1. Grey WF per tonne of crop production for the different management packages.

| Management packages | | | Nitrogen application rate | | | | | | |
|---|---|---|---|---|---|---|---|---|---|
| Fertilizer form | Tillage practice | Irrigation strategy | 25 | 50 | 100 | 150 | 200 | 250 | 300 |
| Nitrate | Conventional | Full irrigation | 108 | 95 | 107 | 122 | 306 | 696 | 1095 |
| Nitrate | Conventional | Deficit irrigation | 90 | 82 | 97 | 138 | 436 | 865 | 1324 |
| Nitrate | No-tillage | Full irrigation | 154 | 136 | 161 | 199 | 294 | 621 | 1002 |
| Nitrate | No-tillage | Deficit irrigation | 139 | 130 | 154 | 203 | 383 | 781 | 1202 |
| Manure | Conventional | Full irrigation | 100 | 83 | 90 | 106 | 167 | 445 | 832 |
| Manure | Conventional | Deficit irrigation | 91 | 75 | 84 | 114 | 231 | 600 | 1028 |
| Manure | No-tillage | Full irrigation | 148 | 121 | 141 | 170 | 221 | 397 | 754 |
| Manure | No-tillage | Deficit irrigation | 134 | 114 | 126 | 168 | 261 | 534 | 927 |



Table A.2. N leaching-runoff potential scores for environmental factors and agricultural practices, following the tier-1
approach (Franke et al., 2013).

| Factors | | | Weight | | Score | Remark |
|---|---|---|---|---|---|---|
| | | | α | β | (s) | |
| Environmental factors | Atmospheric | N-deposition | 10 | 10 | 0 | RFN=0.34 g m$^{-2}$y$^{-1}$ less than 0.5 |
| | Soil | Texture (for leaching) | 15 | 15 | 0.67 | Loam soil |
| | | Texture (for runoff) | 10 | 10 | 0.33 | Loam soil |
| | | Natural drainage (for leaching) | 10 | 15 | 0.67 | Assumed well drained |
| | | Natural drainage (for runoff) | 5 | 10 | 0.33 | Assumed well drained |
| | Climate | Precipitation (mm) | 15 | 15 | 0 | 0-600 very low precipitation (450mm) |
| | N-fixation (kg ha$^{-1}$) | | 10 | 10 | 0 | Non legume crops |
| Agricultural practices | Application rate | | 10 | 0 | * | |
| | Plant uptake (crop yield) | | 5 | 0 | * | |
| | Management practice | | 10 | 15 | 0.33 | Assumed good management practices |

* See Table A.3.
Table A.3. N leaching-runoff potential scores based on fertilizer application rate and plant uptake, and calculated α and β
values following the tier-1 approach.

| Fertilizer application kg ha$^{-1}$ | Categorized | Score for application rate | Score for plant uptake | Calculated α and β | |
|---|---|---|---|---|---|
| | | | | α | β |
| 25 | Very low | 0 | 1 | 0.08 | 0.308 |
| 50 | Low | 0.33 | 0.67 | 0.09 | 0.308 |
| 100 | Low | 0.33 | 0.67 | 0.09 | 0.308 |
| 150 | High | 0.67 | 0.33 | 0.09 | 0.308 |
| 200 | High | 0.67 | 0.33 | 0.09 | 0.308 |
| 250 | Very high | 1 | 0 | 0.09 | 0.308 |
| 300 | Very high | 1 | 0 | 0.09 | 0.308 |
