# Peer review of "Grey water footprint reduction in irrigated crop production: effect of nitrogen application"

_Hydrology and Earth System Sciences, 2017_

## Referee Comment (RC1) · Anonymous Referee #1 · 26 May 2017

The authors make an assessment of the grey and total water footprints of irrigated maize grown in Badajoz, Spain. They use the APEX model to study the effects of 56 management packages to determine the options giving the highest yields and the lowest grey and total water footprints.

I think the subject is interesting for its application to agricultural managements, (after still may improvements) possibly ending in recommendations to agricultural stakeholders in order to decrease water consumption, improve water quality and increase crop yield. The authors have made a full exploration of results based on the results given by

the APEX model.

However, as it is now, the manuscript has more drawbacks than qualities. The problems are the following:

1. Presentation: The language at the beginning is of considerably low quality. Although it improves along the manuscript, the sloppy writing of the introduction, methods and beginning of results puts off the reader. I would recommend improving sentence structure, grammar, term usage, etc, with a professional service. I mention at the end some examples.

2. Site description, Methods. Incredibly the only information of the study site is packed in three words, Spain, maize and Badajoz. Where is this? What are the hydroclimatic characteristics (precipitation, temperature, PET, relative humidity, soil moisture content, water stress), any map? size of the plot, water source, time period of study, elevation, etc. This contrasts with the huge explanation on the parametrization of the APEX model.

3. I know that water foot printing models/ET estimate models on land cover climatic information are not generally calibrated or validated hydrologically. Such appears to be the case of APEX. Although this drawback is well known, the authors do not justify why they are omitting any effort to do so . At least some effort should be done in the manuscript to perform a hydrologic (and/or nutrient load) calibration/validation of APEX in this region, or at least mention and justify why this is impossible to do. Worst case, a good sensitivity analysis of the main parameters regulating the water and N fluxes and/or exhaustive literature review of similar studies shedding some light on the initial parametrization of the model should be included.

4. Does the APEX give an opportunity to choose the PET model? Is Penman-Monteith adequate for this region? Recent studies have found that this model over predicts PET [Milly and Dunne, 2016]. What parameters did you put into Penman Monteith if you didn't have any data?

Milly, P. C. D., and K. A. Dunne (2016), Potential evapotranspiration and continental drying, Nat. Clim. Change, 6(10), 946–949, doi:10.1038/nclimate3046.

4. Based on points 2, 3 and 4, how can you tell which of Tier 1 and Tier 3-APEX is better if you really don't know how accurate are both options due to the lack of observations and real data or calibration or validation? As you state in 489, "the precise values presented here should be taken with caution" and "the outcomes are subject to uncertainties inherent to any modelling effort". This makes me wonder on the real point of reading the manuscript.

Other issues: L. 36-37. First sentence is the worst of all the manuscript. Check language. L. 42 - three quarters of what? L. 66- tillage pan formation? L. 66- no-tillage develops mulch cover? L. 49- Application rate , form of N applied are not practices. L. 50-52 This does not make sense L. 75-79 and and and or or or L. 96 what is a systematic model-based assessment? L. 103 is this really more advanced? in what way? L. 103-104 mention the ties in this sentence first. L. 109 approach applying an approach L. 99-101 Bad English L. 114. I don't think you can determine the added value as it is now. L. 127 "are" partitioned L. 130 Quick and slow component? L. 126-136 It sounds to me as you are just putting in words the ticks/options and numbers that you are entering in the fields of the model. L. 138-145 This is not necessary. Figure 1 has some strange arrows going nowhere. What is a unit of heat accumulation? L. 201 or to surface water through runoff? L. 192-195 Isn't this the main objective of the article? L. 204 Is alpha< or > than beta? L. 212 Eqs. 2 and 3? L. 219 what? L. 229 full irrigation? L. 236 derogation? and check units L. 287 why is it important to be zero? L. 338-352 Isn't this a discussion? Fig. 4 The definition of the three region seems a little bit arbitrary? Why do you put some much emphasis in Region 1 if it is almost the same for all packages? Considering the uncertainty of the analysis I would assume the are really no differences. Figure 6. Nothing makes sense in this figure. Check axis and data on grey and consumptive WF. Or is the difference in magnitude due to green water consumption? Is GW consumption so big in Spain? I don't think so. Everything

here needs explanation. .....

---

## Referee Comment (RC2) · A.-P. Witmer (Referee) · 27 Jun 2017

This paper conforms to the literature regarding virtual water transfers, though it allows me to raise a continuing concern regarding the classification of gre y water footprint (WF) as an absolute, given its abstract dependency on time and location. The modification of environmental regulations by a governmental unit can result in significant differences for embodiment of virtual grey water in an agricultural product, making global water movement tabulation chimerical. Noting this objection, we proceed with review of the paper and its findings. I'm uncomfortable with evaluating the WF in terms only of

Nitrogen, since nitrogen-only inorganic fertilizers significantly affect soil pH. Phosphorus is prevalent in many inorganic fertilizers and in many locations is viewed to have a greater impact on receiving waters than N, thus governing grey WF. Incorporation of P into grey water analysis, or alternatively addressing pH imbalances in N-only fertilizers, could significantly alter the outcome of comparison between manufactured and organic fertilizer impact on WF, and this at least should be acknowledged in the paper. Line 276 – knowing the complexity of Penman-Monteith calculations and the parameters associated with the equation, I'd want to look more closely at data before accepting reference ET calculation for this evaluation. Line 283 – use of zero pest stress impact seems odd for this evaluation. If zero-stress conditions are used, it would make sense to conduct at least a handful of scenarios with high-stress conditions to evaluate the variability of impact based on more extreme ambient states. Discussion/Conclusion – It would be helpful to identify and analyze optimal conditions in terms of balancing grey WF and yield. Can you determine the conditions that generate the best outcome, evaluate them in APEX, and provide data to confirm?

---

## Author Comment (AC1) · 30 Jul 2017

**Reply to Anonymous Referee #1**
We thank Referee #1 for the comments; below we give the reply.

**Comment**
The authors make an assessment of the grey and total water footprints of irrigated maize grown in Badajoz, Spain. They use the APEX model to study the effects of 56 management packages to determine the options giving the highest yields and the lowest grey and total water footprints.

I think the subject is interesting for its application to agricultural managements, (after still may improvements) possibly ending in recommendations to agricultural stakeholders in order to decrease water consumption, improve water quality and increase crop yield. The authors have made a full exploration of results based on the results given by the APEX model.
However, as it is now, the manuscript has more drawbacks than qualities. The problems are the following:

**1. Presentation: The language at the beginning is of considerably low quality. Although it improves along the manuscript, the sloppy writing of the introduction, methods and beginning of results puts off the reader. I would recommend improving sentence structure, grammar, term usage, etc, with a professional service. I mention at the end some examples.**

**Reply:**
We will improve the language of the manuscript at the beginning, with a focus to the introduction section, we will also incorporate the corrections that the referee mentioned as examples.

**Comment**
**2. Site description, Methods. Incredibly the only information of the study site is packed in three words, Spain, maize and Badajoz. Where is this? What are the hydroclimatic characteristics (precipitation, temperature, PET, relative humidity, soil moisture content, water stress), any map? size of the plot, water source, time period of study, elevation, etc. This contrasts with the huge explanation on the parametrization of the APEX model.**
**Reply:**
We agree with reviewer's comment that we did not give enough description of the study area. In fact in our study we want to show the potential for grey WF reduction in a water-scarce area by experimenting the effect of different field-management packages on the grey water footprint of growing crops. As example we used a real agro-hydrologic system in arid environment in water scarce region, which is Badajoz in Spain that is situated in water scarce Guadiana river basin. We will add the following relevant description of the case study area in the data section and in the appendix of the revised version of the manuscript.

The model experiments was carried out for semi-arid climate at Badajoz in Spain (38.88⁰ N, -6.83⁰ E; 185 m above mean sea level). The study area is situated in Guadiana river base, which faces water scarcity during part of the year particularly in summer when water is needed for irrigation (Hoekstra et al., 2012). We run APEX for 20 years (1993-2012) using daily climatic data that includes precipitation, minimum temperature and maximum temperature extracted from the European Climate Assessment and Dataset (Klein Tank et al., 2002). We also used monthly average climatic data such as solar radiation, relative humidity and wind speed from the FAO CLIMAWAT database (Smith, 1993). Daily reference

evapotranspiration is calculated using the Penman-Montheith equation, as implemented in APEX (Williams et al., 2008). The average monthly climatic data are tabulated in Appendix A-4.

Table A.4. The average monthly climatic data of Badajoz in Spain (38.88$^o$ N, -6.83$^o$ E; 185 m above 272 mean sea level), this table will be added in the appendix section of the manuscript.

| Climatic variables | Jan | Feb | Mar | Apr | May | Jun | Jul | Aug | Sep | Oct | Nov | Dec |
|---|---|---|---|---|---|---|---|---|---|---|---|---|
| Temperature max, $^0$C | 14.1 | 16.5 | 20.4 | 22.2 | 26.1 | 31.9 | 34.9 | 34.7 | 30.0 | 24.4 | 18.0 | 14.3 |
| Temperature min, $^0$C | 3.6 | 4.2 | 6.7 | 9.0 | 12.2 | 15.8 | 17.3 | 17.6 | 15.2 | 11.9 | 7.3 | 4.9 |
| Precipitation, mm | 50.2 | 39.5 | 30.9 | 41.1 | 41.9 | 10.8 | 2.3 | 4.2 | 25.1 | 64.4 | 65.2 | 64.0 |
| Solar radiation, MJ/M$^2$ | 7.4 | 10.5 | 12.9 | 19 | 21.9 | 25.7 | 26.9 | 23.9 | 17.8 | 12.3 | 8.1 | 6.4 |
| Relative humidity, % | 83 | 71 | 63 | 56 | 45 | 42 | 37 | 35 | 46 | 64 | 76 | 80 |
| Wind Speed, m/s | 1.7 | 1.9 | 2.09 | 2.09 | 2.2 | 2.3 | 2.4 | 2.2 | 1.81 | 1.6 | 1.7 | 1.7 |
| ET0, mm (penman monteth in APEX) | 33.2 | 57.1 | 108.8 | 145.3 | 196.6 | 224.2 | 250.9 | 218.2 | 139.7 | 83.7 | 43.3 | 29.3 |

The physical and chemical characteristics of the loam soil, and nutrient content in the soil (nitrogen, phosphorus, carbon) are extracted from the 1×1 km$^2$ resolution European Soil Database (Hannam et al., 2009).

Soil moisture content is initialised using the standard procedure in APEX, which is based on average annual rainfall within the period considered (1993-2012). We adjust initial organic-N content for each simulation so that the N build-up in the soil over the 20-year period is zero. We apply the graphical time-series inspection method (Robinson, 2002) to determine the warm-up period, i.e. the period in which simulation results are still affected by the model initialization. We find that we best exclude the first five years of the simulation, thus we show results for the period 1998-2012.

**Comment**
**3. I know that water foot printing models/ET estimate models on land cover climatic information are not generally calibrated or validated hydrologically. Such appears to be the case of APEX. Although this drawback is well known, the authors do not justify why they are omitting any effort to do so. At least some effort should be done in the manuscript to perform a hydrologic (and/or nutrient load) calibration/validation of APEX in this region, or at least mention and justify why this is impossible to do. Worst case, a good sensitivity analysis of the main parameters regulating the water and N fluxes and/or exhaustive literature review of similar studies shedding some light on the initial parametrization of the model should be included.**

**Reply:**
We thank the referee for understanding the data limitation for calibrating and validating the APEX model, which is more true when the experiment is by changing large field-management practices. We put effort to validate our simulation results with earlier studies for N-response curve. As we explained in the manuscript L.467-477, the shape of the N-response curves of our study is comparable with the N-response curve constructed for crops, including maize, for the EU based on field measurements (Godard et al., 2008). Our N-response is also consistent with the results presented by Berenguer et al. (2009), who carried out field experiments for maize for similar conditions in Spain.

In fact it would have been better to calibrate and validate the model for water- and nutrient fluxes; in the revised manuscript we will add a justification why we could not calibrate or validate the hydrologic and nutrient fluxes, also the need of doing it in the subsequent studies.

**Comment**

**4. Does the APEX give an opportunity to choose the PET model? Is Penman-Monteith adequate for this region? Recent studies have found that this model over predicts PET [Milly and Dunne, 2016]. What parameters did you put into Penman Monteith if you didn't have any data?**

Milly, P. C. D., and K. A. Dunne (2016), Potential evapotranspiration and continental drying, Nat. Clim. Change, 6(10), 946–949, doi:10.1038/nclimate3046.

**Reply:**

APEX gives five options to estimate PET: Penman-Monteith, Penman, Priestley Taylor, Hargreaves, and Baier Robertson. In our study we applied Penman Monteith, which is the default method in the model. We have all the required input data to apply Penman Monteith. Though Penman Monteith is commonly used for PET estimation, we find the study by Milly and Dunne (2016) to be relevant; and in the revised version we will add their disclaim on the Penman-Monteith method 'the method over estimate PET as it does not consider the stomatal conductance reductions, which is commonly induced by increasing atmospheric $CO_2$ concentrations)'.

**Comment**

**4. Based on points 2, 3 and 4, how can you tell which of Tier 1 and Tier 3-APEX is better if you really don't know how accurate are both options due to the lack of observations and real data or calibration or validation? As you state in 489, "the precise values presented here should be taken with caution" and "the outcomes are subject to uncertainties inherent to any modelling effort". This makes me wonder on the real point of reading the manuscript.**

**Reply:**

We argue that the comparison of Tier-1 and Tier-3 in the study is still valid as the change in the field-management packages was experimented for the same, default, model parameters. In addition the alpha and beta calculated based on tier-1 level, which is less accurate but easy to estimate the load to freshwater (Franke et al., 2013), does not respond as expected to the changes in the field-management options.

We simulate our experiment using the default parameter in APEX, without calibrating it; and we validate the result based on the N-response curve. We still acknowledge validating of APEX for the water and nutrient fluxes would have increased our confidence to the simulated results, and we will reflect on this in the revised version of the manuscript, also the need of doing it in the subsequent studies.

**Other issues**: (the following comments will be incorporated in the revised article)

**L. 36-37. First sentence is the worst of all the manuscript. Check language**

**L. 42 - three quarters of what?**

      The grey WF from global crop production makes three quarters of the total N-related grey WF in the world (Mekonnen and Hoekstra, 2015).

**L. 66- tillage pan formation?**

      Tillage-pan formation is a formation of compacted soil layer caused by repeated ploughing using heavy weight tillage machineries (Podder et al., 2012).

**L. 66- no-tillage develops mulch cover?**

      By practicing no-tillage the crop residue remains untouched as soil cover, which serves as mulch.

**L. 49- Application rate, form of N applied are not practices.**

Agricultural management practices that influence the grey WF include the N-application rate, the form of N-applied (particularly inorganic-N versus manure or organic-N), and the tillage and irrigation practice.

**L. 50-52 This does not make sense**

A low N-application rate will hamper cop growth and reduce crop yield (Raun et al., 2002). In addition, the low N-application rate will have small water-pollution per hectare, but will have large pollution relative to the amount of crops produced.

**L. 75-79 and and and or or or**

**L. 96 what is a systematic model-based assessment?**

It is an assessment using model in systematic way, which is methodical or a well ordered and efficient way.

**L. 103 is this really more advanced? in what way?**

In this paper the APEX model, process based water- and nitrogen balance and crop growth model, was applied to estimate the grey WF of crop production by tracking the pollutant load to surface water and groundwater with a daily time step. In the previous studies, the pollutant load to surface water and groundwater were estimated based on an annual mass balance approach (Mekonnen and Hoekstra, 2015;Liu et al., 2012). The earlier studies ignore soil organic matter build-up and decomposition, and nitrogen transformations such as mineralization, immobilization and nitrification, which all affect the N uptake and N load to freshwater.

**L. 103-104 mention the tiers in this sentence first.**

Franke et al. (2013) distinguish three tiers, which are ordered 1 to 3 in the increase of accuracy and decrease of feasibility (and data requirement) to estimate the load to freshwater.

**L. 109 approach applying an approach**

The more advanced tier-2 for estimating grey WFs from diffuse pollution is based on an N balance approach, applying a simplified model approach (see for example Mekonnen and Hoekstra (2015), and Liu et al. (2012)).

**L. 99-101 Bad English**

Will be replaced by 'We simulate irrigated-maize growth for twenty-years (1993-2012) at Badajoz in Spain on loam soil in a semi-arid environment'.

**L. 114. I don't think you can determine the added value as it is now.**
**L. 127 "are" partitioned**
**L. 130 Quick and slow component?**

Lateral flow is divided in to two: quick lateral flow joins to the surface runoff quickly; slow later flow components flows as subsurface lateral flow horizontally.

**L. 126-136 It sounds to me as you are just putting in words the ticks/options and numbers that you are entering in the fields of the model.**

**L. 138-145 This is not necessary. Figure 1 has some strange arrows going nowhere. What is a unit of heat accumulation?**

The unit of heat accumulation is growing degree days (GDD).

**L. 201 or to surface water through runoff?**
    or to surface water with runoff

**L. 192-195 Isn't this the main objective of the article?**
    L.192-195 is not the main objective. The main objective of this study is to explore the effect of nitrogen application rate, nitrogen form, tillage practice and irrigation strategy on the nitrogen load to groundwater and surface water, crop yield and the grey water footprint of crop production by a systematic model-based assessment.

**L. 204 Is alpha< or > than beta?**
    Alpha is less than beta

**L. 212 Eqs. 2 and 3?**
    At the tier-1 level, α and β can be estimated using equation 4 and 5 following the guidelines of Franke et al. (2013).

**L. 219 what?**
    where $s_i$ is score for the leaching runoff potential for environmental or management factor i, and $w_i$ is the weight of that factor.

**L. 229 full irrigation?**
**L. 236 derogation? and check units**
    derogation means an exemption from or relaxation of a rule or law. The unit will be corrected to 250 kg N ha$^{-1}$ y$^{-1}$.

**L. 287 why is it important to be zero?**
    N build-up is made zero to avoid N depletion and N surplus in the soil.

**L. 338-352 Isn't this a discussion? Fig. 4 The definition of the three region seems a little bit arbitrary? Why do you put some much emphasis in Region 1 if it is almost the same for all packages? Considering the uncertainty of the analysis I would assume the are really no differences. Figure 6. Nothing makes sense in this figure. Check axis and data on grey and consumptive WF. Or is the difference in magnitude due to green water consumption? Is GW consumption so big in Spain? I don't think so. Everything here needs explanation. .....**

    L 338-352 is meant to explain the result based on the underlining drivers and the processes.

    The definition of the three region in Fig. 4 is not arbitrary. The three regions has unique management package that gives the smallest grey WF, the grey WF in region-I is the smallest with Ma-CT-DI (Manure-conventional tillage and deficit irrigation), the grey WF in region-II is the smallest with Ma-CT-FI, and in region-III with Ma-NT-FI.

    Region-I is shown magnified to add visibility that the grey WF is the smallest for all management packages at N-application rate equal to 50 kg N ha$^{-1}$ y$^{-1}$.

Figure 6 shows the potential change to the grey WF and consumptive WF, if the reference management package is replaced with a management package reduces the total WF.

Figure 6 will be explained in the revised version.

**References:**

Berenguer, P., Santiveri, F., Boixadera, J., and Lloveras, J.: Nitrogen fertilisation of irrigated maize under Mediterranean conditions, European Journal of Agronomy, 30, 163-171, 2009.

Franke, N., Boyacioglu, H., and Hoekstra, A.: Grey water footprint accounting: Tier 1 supporting guidelines, Value of Water Research Report Series No. 65, UNESCO-IHE, Delft, the Netherlands, 2013.

Godard, C., Roger-Estrade, J., Jayet, P.-A., Brisson, N., and Le Bas, C.: Use of available information at a European level to construct crop nitrogen response curves for the regions of the EU, Agricultural Systems, 97, 68-82, 2008.

Hoekstra, A. Y., Mekonnen, M. M., Chapagain, A. K., Mathews, R. E., and Richter, B. D.: Global monthly water scarcity: blue water footprints versus blue water availability, Plos One, 7, e32688, 2012.

Klein Tank, A., Wijngaard, J., Können, G., Böhm, R., Demarée, G., Gocheva, A., Mileta, M., Pashiardis, S., Hejkrlik, L., and Kern-Hansen, C.: Daily dataset of 20th-century surface air temperature and precipitation series for the European Climate Assessment, International journal of climatology, 22, 1441-1453, 2002.

Liu, C., Kroeze, C., Hoekstra, A. Y., and Gerbens-Leenes, W.: Past and future trends in grey water footprints of anthropogenic nitrogen and phosphorus inputs to major world rivers, Ecological Indicators, 18, 42-49, 2012.

Mekonnen, M. M., and Hoekstra, A. Y.: Global gray water footprint and water pollution levels related to anthropogenic nitrogen loads to fresh water, Environmental science & technology, 49, 12860-12868, 2015.

Milly, P. C., and Dunne, K. A.: Potential evapotranspiration and continental drying, Nature Climate Change, 6, 946-949, 2016.

Podder, M., Akter, M., Saifullah, A., and Roy, S.: Impacts of plough pan on physical and chemical properties of soil, Journal of Environmental Science and Natural Resources, 5, 289-294, 2012.

Raun, W. R., Solie, J. B., Johnson, G. V., Stone, M. L., Mullen, R. W., Freeman, K. W., Thomason, W. E., and Lukina, E. V.: Improving nitrogen use efficiency in cereal grain production with optical sensing and variable rate application, Agron J, 94, 815-820, 2002.

Robinson, S.: New simulation output analysis techniques: a statistical process control approach for estimating the warm-up period, Proceedings of the 34th conference on Winter simulation: exploring new frontiers, 2002, 439-446,

Smith, M.: CLIMWAT for CROPWAT. A climatic database for irrigation planning and management, FAO, 1993.

Williams, J. R., Izaurralde, R. C., and Steglich, E. M.: Agricultural Policy/Environmental eXtender Model: Theoretical documentation version 0604, BREC Report, 17, 2008.

---

## Author Comment (AC2) · 30 Jul 2017

**Reply to Ann-Perry Witmer**
We thank Ann-Perry Witmer for the comments; below we give our reply.

**Comment**
This paper conforms to the literature regarding virtual water transfers, though it allows me to raise a continuing concern regarding the classification of grey water footprint (WF) as an absolute, given its abstract dependency on time and location. The modification of environmental regulations by a governmental unit can result in significant differences for embodiment of virtual grey water in an agricultural product, making global water movement tabulation chimerical. Noting this objection, we proceed with review of the paper and its findings.

I'm uncomfortable with evaluating the WF in terms only of Nitrogen, since nitrogen-only inorganic fertilizers significantly affect soil pH. Phosphorus is prevalent in many inorganic fertilizers and in many locations is viewed to have a greater impact on receiving waters than N, thus governing grey WF. Incorporation of P into grey water analysis, or alternatively addressing pH imbalances in N-only fertilizers, could significantly alter the outcome of comparison between manufactured and organic fertilizer impact on WF, and this at least should be acknowledged in the paper.

**Reply:**
Grey WF of growing crop is an indicator of water pollution associated with crop production, it is expressed as the volume of water required to assimilate the pollutant load to meet agreed water quality standards (Hoekstra et al., 2011). If there is modification of environmental regulations by a governmental unit that may change the maximum acceptable concentration of the pollutant load to surface water and groundwater, the calculated volume of grey WF can alter; therefore it is recommended to report the grey WF values with the standards, also with spatial and temporal explicit.

We agree with the referee's concern on the importance of including the grey WF estimation associated with phosphorus (P) as well, particularly in areas where P is a serious threat to the quality of receiving water. In our study we simulate fertilizer application that has not only nitrogen but also nutrients such as phosphorus (P) and potassium (K). While the N-application rates is varying, we always keep P-application rates optimal, that is why we focus presenting the effects of management practices on N-related grey WF.

The grey WF of growing crop associated with the nutrients in fertilizer such as phosphorus, and nitrogen can be estimated, and by definition the nutrient load that requires larger volume of water to assimilate its pollutant load (thus governing grey WF) is reported. In the revised manuscript, we will acknowledge the need to incorporate the P-related grey WF analysis, which will give the overall N-related and P-related grey WF of fertilizer application.

**Comment**
Line 276 – knowing the complexity of Penman-Monteith calculations and the parameters associated with the equation, I'd want to look more closely at data before accepting reference ET calculation for this evaluation.
**Reply:**

We apply Penman-Monteith to calculate the reference ET. As input, we use daily climatic data such as precipitation, minimum temperature and maximum temperature extracted from the European Climate

Assessment and Dataset (Klein Tank et al., 2002). In addition we use monthly average climatic data such as solar radiation, relative humidity and wind speed from the FAO CLIMAWAT database (Smith, 1993). The average monthly values of the input climatic data (minimum and maximum temperature, precipitation, solar radiation, relative humidity, wind speed) and the calculated reference ET will be incorporated in a table in the Appendix of the revised manuscript.

**Comment**
Line 283 – use of zero pest stress impact seems odd for this evaluation. If zero-stress conditions are used, it would make sense to conduct at least a handful of scenarios with high-stress conditions to evaluate the variability of impact based on more extreme ambient states.
**Reply:**
The zero-stress in line 283 is meant for stresses related to weed, pest and diseases in affecting crop growth. Otherwise the effect on crop growth due to other stresses such as stresses from both excess and limitation of water, from limitation of nitrogen, and from very high or very low temperature are simulated.

**Comment**
Discussion/Conclusion
– It would be helpful to identify and analyse optimal conditions in terms of balancing grey WF and yield. Can you determine the conditions that generate the best outcome, evaluate them in APEX, and provide data to confirm?
**Reply:**
As it is shown in Table 2 in the manuscript, given the management practices considered the grey WF and crop yield are best at different N-application rates: grey WF is best (the smallest) at 50 kg N ha$^{-1}$ y$^{-1}$ when yield is not best (small), and crop yield is best (maximum) at 200 kg N ha$^{-1}$ y$^{-1}$ when the grey WF is large. Though the trade-off between improving crop yield and improving grey WF is apparent, the authors share the referees speculation that there would be a conditions that generate optimal for both grey WF and crop yield; exploring these conditions in the study has setbacks mainly from the management options in the model, also this is beyond the scope of the current study.

**References:**

Hoekstra, A. Y., Chapagain, A. K., Aldaya, M. M., and Mekonnen, M. M.: The Water Footprint Assessment Manual: Setting the Global Standard, Earthscan, London, UK, 2011.
Klein Tank, A., Wijngaard, J., Können, G., Böhm, R., Demarée, G., Gocheva, A., Mileta, M., Pashiardis, S., Hejkrlik, L., and Kern-Hansen, C.: Daily dataset of 20th-century surface air temperature and precipitation series for the European Climate Assessment, International journal of climatology, 22, 1441-1453, 2002.
Smith, M.: CLIMWAT for CROPWAT. A climatic database for irrigation planning and management, FAO, 1993.

---

## Referee Report (RR1)

Review
HESS #2017-224
"Grey water footprint reduction in irrigated crop production: effect of N application rate, N form, tillage practice and irrigation strategy"

Summary

In the present study, the authors explore the extent to which fertilizer use and other agricultural management practices impact the agricultural Grey Water Footprint (GWF). The results of the study show that the GWF, which is defined as the amount of freshwater required to assimilate pollutants to meet specific water quality standards, can be reduced by 91% with a decrease in fertilizer application to corn to 50 kg/ha-y. However, with this decrease in the GWF there is also an approximately 70% decrease in yields as well as increase in the blue and green water footprints of crop production. The authors find that the overall water footprint (grey + green + blue) can be minimized with an N application rate of 150 kg/ha-y, with N being applied to the crops as manure and with changes in other management practices (no-till, deficit irrigation). Under this water-optimized scenario, crop yields decrease by 20%. Importantly, the paper shows that there are clear, quantifiable tradeoffs between environmental costs and the human benefits of crop production, and also tradeoffs between avoiding water pollution and reducing water consumption (green and blue water).

**General Comments/Suggestions**

| Lines 49-50 | How does phosphorus play into your assumptions here? Obviously the P content of runoff also contributes to the GWF. In particular, if you are using manure to meet the N requirement, you are (based on typical N:P ratios in manure) almost surely applying excess P, which would increase the GWF of the farming system. You should be explicit about your assumptions here and make clear that, even if P pollution is outside the scope of your analysis, you are likely underestimating the GWF under these scenarios. |
|---|---|
| Lines 49-73 | Very nice discussion of the complex tradeoffs among various agricultural management practices. |
| Lines 94-96 | One of the strengths of the paper is that you explore tradeoffs associated with optimizing for multiple objectives (grey water footprint, blue/green water footprint, crop yield). I think it would be useful to more explicitly include this larger goal here when you state your objectives. |
| Lines 99-101 | Why is Badajoz, Spain a relevant site for your purposes? |

| | |
|---|---|
| Lines 110-114 | You are using a model-based approach that would also allow you to estimate losses of N to the atmosphere.  Although you are specifically interested in water footprint effects here, it is also worth mentioning that managing N use also has impacts on greenhouse gas production.  Optimizing for minimal pollution to the atmosphere would add additional complexity to your discussion of tradeoff.  Although this is certainly beyond the scope of the present study, it would be valuable to mention this part of the agricultural management puzzle. |
| Figure 1 | The labeling on the figure is messed up—maybe this is just a function of the pdf creation, but please check. |
| Line 205 | How is the beta-value defined here different from the Tier-2 mass balance approach?  It seems that beta is also based on a mass balance. |
| Section 3.2 | What about soil health under the low-fertilizer scenarios?  Do yields decrease over time at low fertilizer application rates? |
| Section 3.2 | How do things vary from year to year?  How do calculated GWF rates vary under different temperatures and rainfall quantities? |
| Figure 2 | This is a good figure, but it might be easier to understand if the bars were all the same height (0-100%), so we could see the proportions of the different fluxes varying under the different application rates.  As it is, it is very difficult to see or understand the flux magnitudes at the lower application rates. |
| Figure 3 | Nice figure—the portrayal of the GWF under the normalized and non-normalized conditions is very useful. |
| Lines 332-333 | It is not clear in your discussion how the use of manure N compares to that of commercial N fertilizer in terms of runoff/leaching.  It has been shown that N leaching is usually greater with manure application (Edmeades, 2003, "Long-term effects of manures and fertilisers on soil productivity and quality: a review"), but your results do not seem to support this.  Do the specific conditions of your simulations have an impact on these results? |
| Line 485 | Your statement here again goes back to the issue of why this study site was selected.  It may not be feasible to represent, in this study, a range of climatic conditions to represent all of Europe.  On the other hand, your model results should provide some information regarding how your estimate of GWF effects could vary across varying climates.  Would the |

GWF be larger or smaller in a more humid climate?  How might the recommended regime for optimizing GWF/BWF/GrWF values change across a distribution of climates and landscapes?  As your stated objective is to explore how management practices and N application rates impact the water footprint (not just the water footprint in a semi-arid region of Spain), it would be beneficial to include some comments regarding the wider applicability of your findings.

---

## Author Response (AR2)

- Our reply to the comments in four referee reports are included.
- All the changes in the manuscript are made in track-change mode.

**Suggestions for revision in Report #1**

I have reviewed the subject manuscript and the comments of two earlier reviewers. The authors provide management strategies for reducing nitrogen leaching, using the grey water footprint as the metric for nitrogen pollution. The estimates of nitrogen leaching in response to management strategies are predicted using an un-calibrated model (APEX). I'm afraid I have to agree with the spirit of the first two reviewers' criticism that it is unlikely that an un-calibrated is unlikely to provide any new insight on the well-known tradeoffs between nitrogen application, nitrogen leaching, and crop yield. While the authors can provide numerical results with the APEX model, as long as it's unclear what the uncertainty in those results are, I don't think the authors' recommendations on optimal management strategies are worthwhile. While exploring various model tiers could be interesting, it seems to me that we could have easily predicted that different results would be found for the simple tier-1 versus the more complex tier 3 approach. In addition, because we don't have any data to calibrate to or any idea of the uncertainty in the two models, I don't think the authors can be prescriptive about when the simple tier 1 approach "suffices and when it doesn't."

*We acknowledge that we did not calibrate the model with field experiment data for the specific site, but it would have been difficult to measure nutrient fluxes for a large range of soil, water and nutrient management strategies. However, we put efforts to validate our simulation results i.e. N-response curve, yield and net irrigation requirement with previous studies. Our simulated N-response curve, yields and net irrigation requirement are consistent with the results presented by Godard et al. (2008), Berenguer et al. (2009) and Chukalla et al. (2015), respectively. Other researchers who compared APEX simulated results with values from field experimental studies in similar conditions as that of our site in Spain also reported that APEX can adequately simulate maize yield and leached nitrate-N. Cavero et al. (2012) show that the leached nitrate-N simulated using APEX is equivalent with the observed value. Berenguer et al. (2009) showed a satisfactory agreement of maize yields simulated using APEX to the maize yield observed in a field experimental study for six N-application rates. We think that such consistency in the APEX model makes the model suitable to compare the leaching-runoff fractions estimated with tier-1 and tier-3 approaches. The result that shows that the tier-1 approach underestimates the leaching-runoff fraction beyond an N application rate of 150 kg ha$^{-1}$ is understandable in that for larger N application rates, N uptake by the crop is stabilizing and N surplus increase exponentially, which should increase the N leaching - runoff fraction beyond the proportion to N application. Being able to estimate grey WFs at tier-3 level is a great advancement compared to the reliance of earlier studies on the tier-1 level approach.*

*Nevertheless, we acknowledge the need of validating APEX simulated results with field experiments, also conduct sensitivity analysis for input data and uncertainty analysis for the APEX model. In the discussion section, we already highlighted in line 516-523 the limitation of the research.*

A few other comments:

Much of the English grammar was improved in the revision, but the whole manuscript still needs to be edited by a person whose first language is English.

Move Table 2 to appendix.

The two panels in Figure 3 seem repetitive.

*Based on the suggestion, we moved Table 2 into an appendix. The two panels in Figure 3 are not repetitive: the one in the left is grey WF in m3 per ton, an indicator of pollution per unit of production, and the panel in the right (grey WF in m3 per ha) is an indicator of pollution per unit area. These are complementary indicators.*

**Suggestions for revision in Report #2**

The authors tried to show the trade-off between crop yield, nutrient application rates, and nutrient loading to downstream waterbodies. However, I do not think the findings are new to the literature (please read Journal of Soil and Water Conservation and other relevant journals). In addition, the methodology including APEX model preparation and leaching-runoff fraction definition is unclear and questionable. Furthermore, the results are confusing, and they are not appropriately presented and explained in the manuscript. My specific comments are provided below.

*We do not claim the finding of the current study is entirely new. Yet, we believe that the current paper gave the first insights on the combined effect N application rate, N form, tillage practice and irrigation strategy on the pollution load per unit of crop (or grey WF per tonne).*

*In the manuscript, the leaching-runoff fraction is defined in two ways. The first definition: the leaching-runoff fraction, called α, is the percentage of the amount of chemical applied to the field as fertilizer that is lost to groundwater through leaching or to surface water through runoff. The second definition: the leaching-runoff fraction, now called β, is the percentage of the amount of 'surplus nitrogen' (= input - crop uptake) in the soil that is transported to groundwater by leaching or to surface water by runoff.*

*In the current research, we describe the APEX model (section 2.1), the grey WF concept (2.2), th definition of leaching-runoff fractions (section 2.3) and simulation set-up (section 2.4). However, we did not describe the entire set of equations to simulate the water and nutrient processes and crop growth, which can be referred in the documentation of APEX (Williams et al., 2008).*

Lines 21 to 22: Does this mean the crop yield decreases from 11.1 ton/ha to 3.7 ton/ha when the N application rate is reduced from 300 kg N/ha to 50 kg N/ha? Is so, I do not think this N application scenario is feasible and realistic since farmers will not apply this practice.

*The yield reduces form 11.1 ton/ha to 3.7 ton/ha when the N application rate is reduced from 300 kg N/ha to 50 kg N/ha. This N application scenario is to show the grey WF reduction potential in crop production, which could be practiced by farmers if they are compensated for their yield loss. But anyway, we don't suggest that this is 'feasible', we just present the results.*

Lines 28 to 30: Applying manure can cause another water quality (phosphorus) issues such as algal bloom and eutrophication.

*It is true that applying manure can cause other water quality (phosphorus) issues such as algal bloom and eutrophication. The grey WF value would have been different if the grey WF of phosphorus or other chemicals in crop production are considered. The focus of the current study is on N-related grey WF, which is highlighted in the revised manuscript (line-96).*

Lines 51 to 52: I do not follow this sentence. I do not understand why water pollution (or nutrient loads) is compared to the volume of crops produced. Is this a meaningful comparison?

*The grey water footprint per tonne (the nutrient load per amount of crop produced) is an indicator that compares the pollution and yield per ha. Hence, the indicator can help to optimise pollution per unit of production.*

Lines 49 to 73: What is the main idea of this paragraph?

*The paragraph is a review of literature on the effect of N-application rate, N form, tillage and irrigation practices on yield, N fluxes and crop growth.*

Lines 180 to 182: I wonder if the authors had a chance to evaluate the applicability of the maximum acceptable N concentration (of 50 N mg/L) and natural concentration (of 0.5 mg N/L) to the study watershed. They must be highly varying depending on many factors.

*We adopted the maximum acceptable N concentration, Cmax, based on the EU nitrate directive; which applies for EU member countries including the case study area (Spain). The natural concentration is site specific, however we considered a conservative value (0.5 mg N/L).*

Lines 204 to 220: I do not understand why the leaching-runoff fractions are less than 1.0. For instance, the N load to freshwater bodies can be greater than the N fertilizer applied or the N applied but not taken up by the plant. Please describe the mathematical relationship between the variables.

*The leaching-runoff fraction, $\alpha = \frac{L}{Appl}$*

*L is pollutant load to freshwater bodies contributed by the amount of chemical applied (Appl) to the field. N is supplied to the soil through many paths including N anthropogenic application, N deposition and N fixation. However, the grey WF is an indicator to measure pollution due to anthropogenic chemical application (Total pollutant minus pollutant load caused by non-anthropogenic addition). Thus L is always less than Appl and the leaching-runoff fraction should be less than 1.*

*The description of L is improved in the manuscript (Line 213): "L (kg N ha-1 y-1) is the N load to freshwater bodies due to the anthropogenic N addition".*

Lines 241 to 253: There is no information about the reliability of the assumptions made. I wonder if the authors had a chance to compare the assumptions with other reference values, statistics, and field observations. Lines 282 to 290: I could not find information about how the other parameters (for flow, N cycle and transport) were determined. I believe they might be calibrated to measurements, but there is no description about the parameter calibration.

*We acknowledge the model was not calibrated with field experiment data. However, efforts have been put to validate our simulation results which is already explained in the reply to referee#1 and also discussed in the manuscript.*

Line 361: Could you explain why the grey WF is minimized at the N application rate of 50 kg N/ha rather than 25 kg N/ha? In addition, the authors might be able to find an N application rate that minimizes grey WF between 50 and 100 kg N/ha. In addition, I wonder if these results can be applicable to other cases and watersheds.

*While increasing the N-application rate from 25 to 50 kg N ha$^{-1}$ ya$^{-1}$, crop yield increases at a slightly faster rate than N pollutant load (the nitrogen efficiency is high at lower N application rates); as a result, the grey WF per tonne slightly drops. At higher N application, the yield increases but at slower rate than the N load and thus the grey WF per tonne will increase.*

*In the manuscript, we reported results of simulation for the N application rates at 25, 50, 100, 150, 200, 250 or 300 kg N ha$^{-1}$y$^{-1}$. We did not simulate for all intermediate N application rates, however the simulation result at*

*N = 75kg N ha$^{-1}$ y$^{-1}$ under the reference management package (inorganic N fertilizer in combination with conventional tillage and full irrigation) is consistent with the results.*

Lines 373 to 392 and Figure 5: In Figure 5b, the largest consumptive WF (blue and green WFs) is found at the lowest nitrogen application rates (of 25 kg N/ha), and it rapidly decreases with increases in the application rates. When nitrogen application rates low, however, crop yields are also low, and ETs are low as well, as demonstrated in Figure 5a. Thus, consumptive WF should be low when the nitrogen application rates are low (rather than high like 3000 m3/ton, compared to 700 m3/ton). Please explain why the consumptive WF is high even though ET is low when nitrogen application rate is low.

*The consumptive WF at low N rate is high (3000 m3/ton) even though ET and yield are low. This is because the reduction in yield from the high N application to the low N application rate is faster (83.7%, 11.1 to 1.8 ton/ha) than the reduction in ET (23.6%, 762 to 582 mm).*

Lines 402 to 447: I do not understand these descriptions and explanations.

*This section describes the results of the side objective of the current study (line 116 - 118): "comparing the N leaching-runoff fractions that result from the APEX simulations with the leaching-runoff fractions estimated with the simpler tier-1 approach, in order to find out the added value of employing the advanced model approach."*

Lines 464 to 464 and Table 2: I believe it is good to show three WFs separately rather than total WF so that the authors can effectively discuss the strengths and weaknesses of each scenario.
Lines 488 to 492: I think the findings mentioned are well known (not new to the literature). The authors just used different terms such as grey, blue, and green water footprint.

*We believe that the current paper is the first to provide insight on the combined effect of N application rate, N form, tillage practice and irrigation strategy on the pollution load per unit of crop (or grey WF per tonne).*

**Suggestions for revision in Report #3**
GENERAL COMMENTS
This manuscript details the results of a modeling exercise to assess the fate of nitrogen and grey water footprint size in agricultural systems under different management practices. The authors use the Agricultural Policy Environmental eXtender (APEX) model to assess crop yields and nutrient leaching/ runoff, and results are compared against a more simplistic, less-data intensive, expert-based approach for estimating N loads from agricultural areas. The assessments are based on climate and agricultural characteristics of a farming region in Spain. Results indicated that for low levels of nitrogen application, the simple approach yields approximately the same results as the more data-intensive modeling endeavor. At higher levels of nitrogen application, the simple method underestimates nitrogen leaching/runoff. The modeling results also explored multiple types and combinations of management practices, finding there are inevitable tradeoffs between crop yield and water pollution, and water pollution and water consumption.

SPECIFIC COMMENTS
This is an interesting paper with potential implications for applied agricultural management to reduce water pollution problems while minimizing negative impacts to crop yields. The following are a few comments that could help improve this work if incorporated.

- Lines 79-84 provide an overview of studies that have explored the impacts of crop yield and N management via various agricultural management practices. However, what would be more helpful to the reader is an overview of those study findings, and how these findings informed the present work.

*The purpose of the section (Line 79-84) is to inform the current work on the number of management packages that are considered in the previous researches. The previous studies are for different environments and sometimes for different crop types than the current case study. Thus the effect of the management package on the yield, water productivity and leaching are not quantified for a comparison with the current study.*

- I realize that validating model results with field experiments is difficult, and the authors clearly explain why this is so (Lines 90-94). I also appreciate the author's comparison of their model results with those of Berenguer et al 2009- particularly given that the APEX results are only really comparing Tier 1 and Tier 3 levels. Given the lack of field data available, this multi-model comparison is very important. However, I feel the discussion and comparison of model results is short and lacks depth. There is brief mention of modeling results based on EU studies, and research by Berenguer et al 2009, but digging in and really discussing the differences and similarities between these results would add greatly to this work.

*In addition to showing the comparison of our study result with the research by Berenguer et al., (2009) in Figure 10; in the revised manuscript, the difference between our study and the study by Berenguer et al., (2009) is quantified to be 25% higher and this is highlighted in Line 489.*

- At the beginning of Section 2.3, the authors state that part of the reason for evaluating Tier 3 vs Tier 1 is to see if "the simpler estimation approach (Tier 1) as applied to previous studies" is an accurate measure. I'm not sure what these "other studies" are in particular, maybe they're the ones I mention in the comment above, but either way, there could be much value added to explicitly showing the comparisons between Tier 1 results and Tier 3 results, if possible?

*The previous studies that applied the Tier 1 approach are the studies by Mekonnen and Hoekstra (2011), and Brueck and Lammel (2016), which are mentioned in Line 110 of the manuscript. Section 3.4., Figures 7 to 9 and Lines 419 to 465 give explicit explanation and comparison of Tier 1 and Tier 3.*

TECHNICAL COMMENTS

- In line 36, "fertilizer" should be "fertilizers".
*Considered.*

- The authors use the spelling "freshwater" and "fresh water" in this manuscript (see lines 38 and 39 for examples). One spelling version should be selected and used consistently throughout.
*Fresh water is now consistently used in the revised manuscript.*

- In line 157, I suspect the authors meant "Phenological", not "Phonological".
*Yes we meant "Phenological", it is corrected.*

- Equations 4 & 5 (Lines 220-223) reference work done by Franke et al 2013. In these equations, there is a variable s¬_i that is a "score for leaching runoff potential", while I realize this is information from another paper, it would be helpful to discuss how these values are developed, even if briefly.

*The following description is added in the revised manuscript in the Line 226 to 229.*
*Corresponding to a certain state of factor, a score s is between 0 and 1: scores of 0, 0.33, 0.67 and 1 refer to a very low, low, high and a very high leaching-runoff potential, respectively. A weight w per factor i denotes the importance of the factor. The weights given to the separate influencing factors add up to a total of 100.*
*These values are developed by the Grey Water Footprint Expert Panel, who are formed to guide on how to best estimate the values of the leaching-runoff fraction at Tier-1 level.*

- In Figs 8 & 9, I assume the "calculated, tier-1" numbers (in blue) are mean values? Please specify in the body of the manuscript for clarity.

*The " calculated leaching-runoff fractions, Tier-1" (in blue) are not mean values; the line looks horizontal and thus the calculated Tier-1 values look equal due to that fact that the difference is small.*

**Suggestions for revision in Report #4**
In the present study, the authors explore the extent to which fertilizer use and other agricultural management practices impact the agricultural Grey Water Footprint (GWF). The results of the study show that the GWF, which is defined as the amount of freshwater required to assimilate pollutants to meet specific water quality standards, can be reduced by 91% with a decrease in fertilizer application to corn to 50 kg/ha-y. However, with this decrease in the GWF there is also an approximately 70% decrease in yields as well as increase in the blue and green water footprints of crop production. The authors find that the overall water footprint (grey + green + blue) can be minimized with an N application rate of 150 kg/ha-y, with N being applied to the crops as manure and with changes in other management practices (no-till, deficit irrigation). Under this water-optimized scenario, crop yields decrease by 20%. Importantly, the paper shows that there are clear, quantifiable tradeoffs between environmental costs and the human benefits of crop production, and also tradeoffs between avoiding water pollution and reducing water consumption (green and blue water).

*General Comments/Suggestions*
Lines 49-50 How does phosphorus play into your assumptions here? Obviously the P content of runoff also contributes to the GWF. In particular, if you are using manure to meet the N requirement, you are (based on typical N:P ratios in manure) almost surely applying excess P, which would increase the GWF of the farming system. You should be explicit about your assumptions here and make clear that, even if P pollution is outside the scope of your analysis, you are likely underestimating the GWF under these scenarios.

*The focus of the current study is on N-related grey WF, which is highlighted in the revised manuscript (line-96). Line 529-531 also reflect on the focus of the current study.*

Lines 49-73 Very nice discussion of the complex tradeoffs among various agricultural management practices. Lines 94-96 One of the strengths of the paper is that you explore tradeoffs associated with optimizing for multiple objectives (grey water footprint, blue/green water footprint, crop yield). I think it would be useful to more explicitly include this larger goal here when you state your objectives.

Lines 99-101 Why is Badajoz, Spain a relevant site for your purposes?

*Badajoz is located in the water-scarce Guadiana river basin, where reducing the green, blue and grey WF has a positive impact in managing water scarcity.*

Lines 110-114 You are using a model-based approach that would also allow you to estimate losses of N to the atmosphere. Although you are specifically interested in water footprint effects here, it is also worth mentioning that managing N use also has impacts on greenhouse gas production. Optimizing for minimal pollution to the atmosphere would add additional complexity to your discussion of tradeoff. Although this is certainly beyond the scope of the present study, it would be valuable to mention this part of the agricultural management puzzle.

*The focus of the current study is highlighted in the objective (introduction section) and in the discussion section.*

Figure 1 The labeling on the figure is messed up—maybe this is just a function of the pdf creation, but please check.

*The comment is addressed in the revised version.*

Line 205 How is the beta-value defined here different from the Tier-2 mass balance approach? It seems that beta is also based on a mass balance.

*The calculation of N load (beta, leaching-runoff fraction) to fresh water in Tier-2 ignores soil organic matter build-up and decomposition, and nitrogen transformations such as mineralization, immobilization and nitrification, which all affect the N uptake and N load to fresh water. We already discussed the comparison between Tier-1, Tier-2 and Tier-3 approaches in lines 105 to 118 of the manuscript.*

Section 3.2 What about soil health under the low-fertilizer scenarios? Do yields decrease over time at low fertilizer application rates?

*Yes, soil fertility decreases under low-fertilizer scenarios due to nutrient mining.*

Section 3.2 How do things vary from year to year? How do calculated GWF rates vary under different temperatures and rainfall quantities?

*The inter-annual grey WF variation is not significant, as irrigation is applied even if the rainfall quantity varies year after year.*

Figure 2 This is a good figure, but it might be easier to understand if the bars were all the same height (0-100%), so we could see the proportions of the different fluxes varying under the different application rates. As it is, it is very difficult to see or understand the flux magnitudes at the lower application rates.

*Presenting Figure 2 with all bars at the same height (in %) can show the proportions of the different fluxes under the different N application rates, however this option hides the magnitude of the total N load to fresh water at different N application rates.*

Figure 3 Nice figure—the portrayal of the GWF under the normalized and nonnormalized conditions is very useful. Lines 332-333 It is not clear in your discussion how the use of manure N compares to that of commercial N fertilizer in terms of runoff/leaching. It has been shown that N leaching is usually greater with manure application (Edmeades, 2003, "Long-term effects of manures and fertilisers on soil productivity and quality: a review"), but your results do not seem to support this. Do the specific conditions of your simulations have an impact on these results?

*In our simulation, N leaching is shown to be greater from N fertilizer than N in manure. This is in line with other studies explained in Line 57 to 61 of the manuscript. Inorganic N is readily available for uptake by crops (Haynes, 2012), whereas the organic-N contained in manure becomes available only gradually, as it should first be*

*converted (mineralized) to inorganic form (Ketterings et al., 2005). The mobile nature of nitrate makes it susceptible for higher risk of leaching (Yanan et al., 1997), while the slow disappearance of manure makes it susceptible to N losses through runoff before being taken up by the crop (Withers and Lord, 2002).*

Line 485 Your statement here again goes back to the issue of why this study site was selected. It may not be feasible to represent, in this study, a range of climatic conditions to represent all of Europe. On the other hand, your model results should provide some information regarding how your estimate of GWF effects could vary across varying climates. Would the GWF be larger or smaller in a more humid climate? How might the recommended regime for optimizing GWF/BWF/GrWF values change across a distribution of climates and landscapes? As your stated objective is to explore how management practices and N application rates impact the water footprint (not just the water footprint in a semiarid region of Spain), it would be beneficial to include some comments regarding the wider applicability of your findings.

*Exploring the validity of the findings at Badajoz at a range of climatic conditions that represent all Europe is interesting, which is however beyond the scope of the current study.*

**References**

[revised manuscript text omitted]